# C²R: Cross-sample Consistency Regularization Mitigates Feature Splitting and Absorption in Sparse Autoencoders

**Haoran Jin** [1 2]  **Xiting Wang** [3]  **Shijie Ren** [3]  **Hong Xie** [1 2]  **Defu Lian** [1 2]

## Abstract

Sparse Autoencoders (SAEs) are widely used to interpret large language models by decomposing activations into sparse, human-understandable features, but scaling to large dictionaries exposes fundamental challenges. Systematic studies reveal pervasive feature splitting that fragments coherent concepts into non-atomic latents and widespread feature absorption that creates arbitrary exceptions in general features, severely compromising latent reliability. These issues stem from inconsistent latent assignment across samples: without cross-sample constraints, per-sample optimization often allows a single underlying concept to be inconsistently distributed across multiple redundant or interfering latents. To address this, we introduce C²R (**C**ross-sample **C**onsistency **R**egularization). C²R explicitly encourages that each semantic feature is consistently represented by a unified latent across the batch by penalizing the co-activation of directionally similar latents. Comprehensive evaluation demonstrates that C²R effectively mitigates both splitting and absorption while, crucially, preserving reconstruction fidelity, providing a principled solution that enhances latent interpretability without degrading model performance. Source code is available[*].

## 1. Introduction

Sparse Autoencoders (SAEs) have emerged as a powerful tool for interpreting large language models (LLMs), breaking down complex internal representations into sparse,

| SAE Constraints | Theoretical Guarantee | | Intuitively Solved | | Reconstruction Preservation |
|---|---|---|---|---|---|
| | Splitting | Absorption | Splitting | Absorption | |
| $\ell_1$ | × | × | × | × | ✓ |
| TopK | × | × | × | × | ✓ |
| Batch TopK | × | × | × | × | ✓ |
| Matryoshka | × | × | ✓ | ✓ | × |
| Ort | × | × | × | ✓ | ✓ |
| Ours | ✓* | ✓* | ✓ | ✓ | ✓ |

*Table 1.* Comparison of different SAE constraints in terms of theoretical guarantees, intuitive solutions, and reconstruction fidelity. Our cross-sample consistency regularization uniquely offers a theoretical guarantee against splitting and absorption and preserves reconstruction fidelity. *Theoretical guarantee holds under the condition in Eq. 13, which is empirically satisfied in 88.1% of absorption pairs (see Appendix K).

interpretable features (Huben et al., 2023; Bricken et al., 2023). These features provide valuable insights into reasoning, alignment (Zhao et al., 2025; Yeo et al., 2025; Wang et al., 2025), knowledge awareness, hallucinations (Ferrando et al.), and cross-model feature spaces (Lan et al., 2025) of LLMs. This approach is grounded in the hypothesis that the semantic *features* in a model's representation are more effectively captured by an overcomplete sparse basis (Olshausen & Field, 1997) than by dense neuron activations, which tend to be polysemantic. Ideally, each latent in an SAE corresponds to a single, human-interpretable concept.

While SAEs are effective, they face significant challenges, specifically feature splitting (Bricken et al., 2023; Leask et al.) and feature absorption (Chanin et al., 2024), which undermine the reliability of learned latents. Feature splitting fragments coherent, high-level concepts into overly specific pieces. For example, a single "Mathematics" feature might break down into separate latents for "Algebra," "Geometry," and others (Chanin et al., 2024). Although these granular features are interpretable, this fragmentation is problematic because it obscures the true high-level concept the model functionally uses. Moreover, this is inefficient: the dictionary wastes capacity on redundant variations of known concepts instead of finding new ones. Feature absorption, on the other hand, creates "holes" in general features when specific latents capture their activations. A "starts with S" latent, for instance, might fail to activate on "short" or "small" because token-specific latents absorb the signal. This effectively changes the latent to "starts with S (except for short/small)," distorting the intended pattern. Leask et al.

[1]University of Science and Technology of China [2]State Key Laboratory of Cognitive Intelligence [3]Gaoling School of Artificial Intelligence, Renmin University of China. Correspondence to: Xiting Wang <xitingwang@ruc.edu.cn>, Hong Xie <hongx87@ustc.edu.cn>, Defu Lian <liandefu@ustc.edu.cn>.

*Proceedings of the 43rd International Conference on Machine Learning*, Seoul, South Korea. PMLR 306, 2026. Copyright 2026 by the author(s).

[*]https://github.com/hr-jin/Cross-sample-Consistency-Regularization

show that splitting scales with model size, while Chanin et al. (2024) find that absorption affects hundreds of LLM SAEs. These systematic failures undermine the utility of SAEs for critical tasks like causal analysis and circuit discovery.

We argue that these failures arise from a mismatch between the hierarchical nature of language model features and the local scope of standard sparsity constraints. Real-world concepts are inherently hierarchical (Bussmann et al.), yet common objectives like $\ell_1$ (Bricken et al., 2023) or TopK (Gao et al.) enforce sparsity on a per-sample basis, which potentially penalizes the hierarchical structure. activating both uses more of the sparsity budget than activating the child feature alone. Consequently, the optimizer suppresses the parent latent and forces the child latent to take over its role to keep the active count low. Similarly, regarding feature splitting, the objective does not distinguish between activating a general latent or a specific one. The SAE allows disjoint latents to handle different contexts of a single concept, as nothing ensures consistent latent assignment across samples. Solving these issues, therefore, requires looking beyond per-sample optimization to enforce cross-sample consistency in how latents are selected.

To address this issue, we propose C²R (**C**ross-sample **C**onsistency **R**egularization). This objective builds on the geometry of the Minkowski inequality (Gruber, 1979) and the strict convexity of the $\ell_2$ norm. It exploits the fact that the sum of the norms of separate vectors strictly exceeds the norm of their sum: $\|u\|_2 + \|v\|_2 > \|u + v\|_2$ for non-aligned vectors. By applying this constraint across the batch dimension, C²R makes it expensive to split a concept into multiple disjoint latents. This formulation penalizes spreading semantic information across redundant latents, driving the SAEs to consolidate activations into a single, consistent latent without supervision.

Our contributions are threefold:

- **Theoretical diagnosis:** We identify the lack of cross-sample consistency in per-sample sparsity objectives as the root cause of feature splitting and absorption, providing a unified formal analysis of these phenomena.

- **Principled objective:** We propose C²R, a novel regularization objective that utilizes decoder geometry and batch-level statistics to enforce consistent latent selection, effectively distinguishing between true polysemanticity and harmful redundancy.

- **Empirical validation:** We demonstrate that C²R significantly mitigates splitting and absorption, achieving better feature hierarchy without compromising reconstruction fidelity compared to state-of-the-art baselines.

## 2. Related Work

### 2.1. Sparse Autoencoders

Sparse Autoencoders (SAEs) are grounded in the linear representation hypothesis, which posits that the dense activation space of a language model is constructed from the superposition of sparse, discernible concepts, referred to as *features*. The goal of an SAE is to recover these ground-truth features by learning a dictionary of *latents*. Ideally, there exists a one-to-one mapping where each learned latent corresponds precisely to a single meaningful feature.

Formally, given an input activation vector $x \in \mathbb{R}^{d_{model}}$ (e.g., from a Transformer's residual stream), an SAE projects $x$ into a higher-dimensional sparse latent code $f \in \mathbb{R}^{d_{dict}}$, where $d_{dict} \gg d_{model}$. The encoding process is parameterized by an encoder weight matrix $W_e \in \mathbb{R}^{d_{dict} \times d_{model}}$ and a bias $b_e$:

$$f = \phi(W_e x + b_e), \tag{1}$$

where $\phi$ is a non-linear activation function, typically ReLU, TopK, or JumpReLU, designed to induce sparsity. The input is then reconstructed via a linear decoder $W_d \in \mathbb{R}^{d_{model} \times d_{dict}}$:

$$\hat{x} = W_d f + b_d. \tag{2}$$

The training objective minimizes a combination of reconstruction error and a sparsity penalty:

$$\mathcal{L}_{\text{SAE}}(x) = \|x - \hat{x}\|_2^2 + \lambda \mathcal{R}(f). \tag{3}$$

Common choices for the regularizer $\mathcal{R}(f)$ include the $\ell_1$ norm (Bricken et al., 2023) or the auxiliary loss associated with TopK constraints (Gao et al.). Various architectural improvements have been proposed to enhance SAE quality. Gated SAEs (Rajamanoharan et al., 2024a) and JumpReLU SAEs (Rajamanoharan et al., 2024b) introduce learnable thresholds to improve the fidelity-sparsity frontier. However, these methods focus on the per-sample activation sparsity rather than enforcing the hierarchical structure of the SAE.

### 2.2. Structural SAEs

More recently, approaches attempting to structure the latent space have emerged. Batch TopK SAEs (Leask et al.) relax the rigid per-sample TopK constraint to a batch-level aggregate, allowing for variable sparsity across samples. While this improves reconstruction, it lacks any mechanism to enforce hierarchical structure among the latents and still faces feature absorption and splitting challenges.

Matryoshka SAEs (Bussmann et al.) enforce a nested structure where subsets of latents are trained to approximate the input at different sparsity levels. While this creates a hierarchy, it compromises reconstruction fidelity and lacks a clear theoretical explanation for why it would fix the optimization issues of standard sparsity objectives. OrtSAE (Korznikov

et al., 2025) addresses feature splitting and composition by penalizing the cosine similarity between decoder weights. However, this approach tries to handle absorption indirectly via the decoder geometry, rather than addressing the encoder activation patterns where absorption is formally defined. In contrast, our method targets the latent activations directly to penalize redundancy, offering a theoretically guaranteed solution derived from the formal definitions of splitting and absorption.

## 2.3. Minkowski Inequality (Gruber, 1979)

For any real number $p \geq 1$ and any two real sequences $a = (a_1, a_2, \ldots, a_n)$ and $b = (b_1, b_2, \ldots, b_n)$, their $p$-norms satisfy

$$\|a + b\|_p \leq \|a\|_p + \|b\|_p. \tag{4}$$

When the activations of a single semantic feature are distributed across multiple redundant SAE latents, the combined $p$-norm of their activations exceeds that of a single latent capturing the same feature. This inequality motivates our regularization term that penalizes redundant feature allocation across latents, thereby constraining feature splitting and absorption.

## 3. Unified Problem Formulation

In this section, we propose a geometric framework that unifies feature splitting and absorption. Rather than viewing them as separate pathologies, we model them as instances of latent redundancy arising from perturbed basis directions. We introduce a **redundancy parameter** $\alpha$ to quantify the extent to which a semantic feature "leaks" into varying latents, allowing us to derive a single consistency condition that prevents both failure modes.

Let $L_1$ denote the ideal latent direction that captures the complete semantic feature $F$, and let $L_2$ denote an orthogonal direction that captures residual, non-feature components. Both $L_1$ and $L_2$ are unit vectors and orthogonal to each other:

$$\|L_1\| = \|L_2\| = 1, \quad L_1 \perp L_2. \tag{5}$$

Let $z_1^{(i)}$ and $z_2^{(i)}$ be the corresponding activations of $L_1$ and $L_2$ for sample $i$. In the ideal case (no splitting nor absorption, $\alpha = 0$), all information related to $F$ is represented solely by $L_1$, and $L_2$ contributes only to the orthogonal reconstruction components. The activation pattern is:

| Sample | $L_1$ | $L_2$ |
|--------|-------|-------|
| 1 | $z_1^{(1)}$ | 0 |
| $\vdots$ | $\vdots$ | $\vdots$ |
| $m$ | $z_1^{(m)}$ | 0 |
| $m+1$ | $z_1^{(m+1)}$ | $z_2^{(m+1)}$ |
| $\vdots$ | $\vdots$ | $\vdots$ |
| $m+n$ | $z_1^{(m+n)}$ | $z_2^{(m+n)}$ |

Here, $z_1^{(i)}$ corresponds to the feature-aligned activation along $L_1$, while $z_2^{(i)}$ encodes the orthogonal residual components required for accurate reconstruction. Samples $1, \ldots, m$ exclusively activate $L_1$, whereas samples $m+1, \ldots, m+n$ possess a non-zero component from $L_2$. In practice, sparse autoencoders often learn perturbed latent directions, denoted $L_1'$ and $L_2'$, that deviate from the ideal basis. We assume $L_2'$ contains a fraction $\alpha \in [0, 1]$ of the feature direction $L_1$, forming a new latent that partially overlaps with it:

$$L_1' = L_1, \quad L_2' = \frac{(1-\alpha)L_1 + \alpha L_2}{\|(1-\alpha)L_1 + \alpha L_2\|}. \tag{6}$$

Here $\alpha$ quantifies the degree of cross-latent feature sharing. When $\alpha = 0$, $L_2'$ is perfectly orthogonal to $L_1'$ (no dispersion). When $\alpha = 1$, $L_2'$ fully aligns with $L_1'$, corresponding to a complete feature split into two disjoint latents.

The corresponding activation pattern becomes:

| Sample | $L_1'$ | $L_2'$ |
|--------|--------|--------|
| 1 | $z_1^{(1)}$ | 0 |
| $\vdots$ | $\vdots$ | $\vdots$ |
| $m$ | $z_1^{(m)}$ | 0 |
| $m+1$ | $(1-\alpha)z_1^{(m+1)}$ | $\sqrt{(\alpha z_1^{(m+1)})^2 + (z_2^{(m+1)})^2}$ |
| $\vdots$ | $\vdots$ | $\vdots$ |
| $m+n$ | $(1-\alpha)z_1^{(m+n)}$ | $\sqrt{(\alpha z_1^{(m+n)})^2 + (z_2^{(m+n)})^2}$ |

*Table 2.* Activation pattern when feature splitting or absorption occurs.

This activation pattern illustrates that for samples $m+1$ through $m+n$, part of the original feature direction $L_1$ is reconstructed via $L_2'$ due to the shared component $\alpha L_1$ in Eq. 6. The $\sqrt{(\alpha z_1^{(i)})^2 + (z_2^{(i)})^2}$ term ensures the total reconstructed model activation remains the same through vector addition of $L_1'$ and $L_2'$. This formalization unifies feature splitting and absorption: splitting and full absorption correspond to $\alpha=1$, where feature-aligned energy is duplicated across latents, and partial absorption corresponds to $\alpha<1$, where the $L_2'$ inherits part of the feature along with its orthogonal component.

## 4. Theoretical Analysis on Two SAE Latents

### 4.1. Limitation of Per-sample Sparsity Objectives

We analyze the behavior of $\ell_1$ and TopK objectives under the activation patterns defined in the unified problem formulation.

**Lemma 4.1.** *Per-sample sparsity constraints, specifically $\ell_1$ regularization and TopK, strictly favor feature splitting and absorption ($\alpha \to 1$) over the ideal orthogonal decomposition ($\alpha = 0$) given equivalent reconstruction fidelity.*

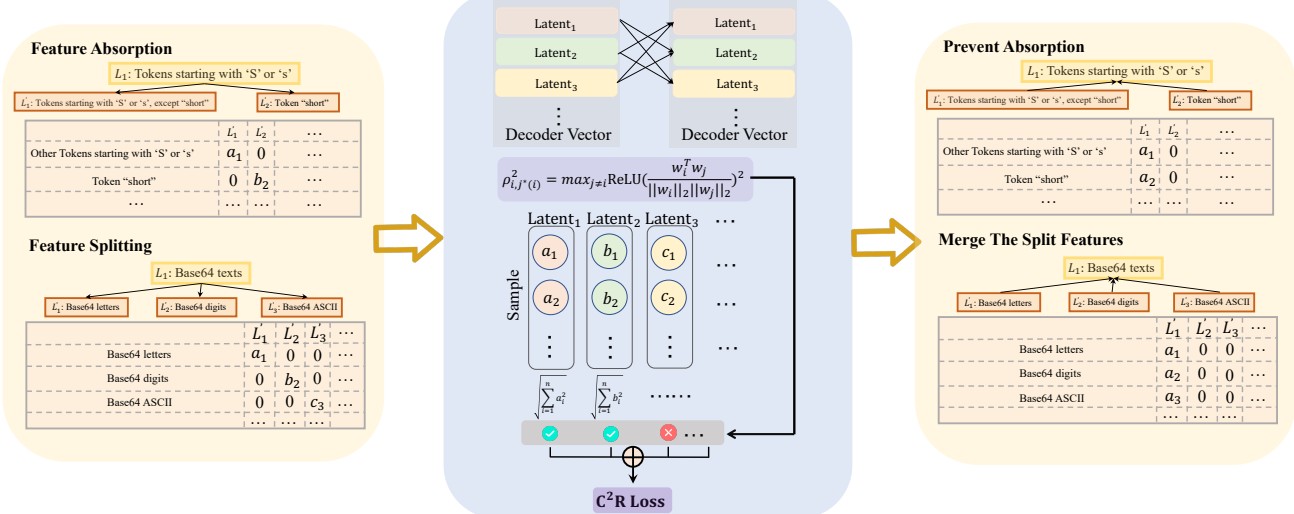

*Figure 1.* Overview of C²R (Cross-sample Consistency Regularization). Each mini-batch contains activations of multiple samples encoded by a sparse autoencoder. C²R enforces consistency of latent usage across samples by constraining activation patterns along the batch dimension. This encourages each latent to represent a complete semantic feature rather than fragmented or absorbed subfeatures, mitigating feature splitting and absorption.

*Proof.* **Case 1: The $\ell_1$ Penalty.** We compare the cumulative $\ell_1$ penalty for the ideal configuration versus the perturbed configuration, assuming equivalent SAE reconstruction.

(1) **Ideal Decomposition ($\alpha = 0$):** In this state, the feature $L_1$ and the residual $L_2$ are orthogonal. The total $\ell_1$ norm over the batch is:

$$\ell_1(\alpha = 0) = \sum_{i=1}^{m+n} z_1^{(i)} + \sum_{i=m+1}^{m+n} z_2^{(i)}. \qquad (7)$$

(2) **Perturbed Decomposition ($\alpha > 0$):** Under splitting or absorption, the activation energy is redistributed. The total $\ell_1$ norm becomes:

$$\ell_1(\alpha > 0) = \sum_{i=1}^{m} z_1^{(i)} + \sum_{i=m+1}^{m+n} (1 - \alpha) z_1^{(i)} \\ + \sum_{i=m+1}^{m+n} \sqrt{(\alpha z_1^{(i)})^2 + (z_2^{(i)})^2}. \qquad (8)$$

We apply the Minkowski inequality to the terms summing over indices $i \in \{m+1, \ldots, m+n\}$. For any $\alpha > 0$:

$$\sum_{i=m+1}^{m+n} \alpha z_1^{(i)} + \sum_{i=m+1}^{m+n} z_2^{(i)} > \sum_{i=m+1}^{m+n} \sqrt{(\alpha z_1^{(i)})^2 + (z_2^{(i)})^2}. \qquad (9)$$

Subtracting the shared terms from Eq. 7 and Eq. 8, it follows that $\ell_1(\alpha > 0) < \ell_1(\alpha = 0)$. The strict inequality holds for all $\alpha \in (0, 1]$. Consequently, the optimization process minimizes the global objective by maximizing $\alpha$, thereby

driving the solution toward splitting or absorption to reduce the total norm while maintaining reconstruction fidelity.

**Case 2: The TopK Constraint.** The TopK objective enforces a hard constraint on the number of active latents, effectively minimizing the $\ell_0$ norm of the activation vector for a fixed reconstruction error tolerance. We evaluate the cardinality of the active set for the intersection samples $i \in \{m+1, \ldots, m+n\}$.

(1) **Ideal Decomposition ($\alpha = 0$):** The signal consists of two orthogonal non-zero components: the feature activation $z_1^{(i)}$ and the residual $z_2^{(i)}$. Since the basis vectors $L_1$ and $L_2$ are orthogonal, exact representation requires both to be active. Thus, the sparsity consumption is:

$$\|z^{(i)}\|_0 = 2 \quad \text{for } i \in \{m+1, \ldots, m+n\}. \qquad (10)$$

(2) **Full Splitting or Absorption ($\alpha = 1$):** Substituting $\alpha = 1$ into the activation pattern defined in Table 2, the coefficient for the first latent becomes zero: $(1 - \alpha) z_1^{(i)} = 0$. The second latent, $L_2'$, captures the entire vector magnitude $\sqrt{(z_1^{(i)})^2 + (z_2^{(i)})^2}$. As a result, the SAE represents the same vector space using a single active latent:

$$\|z^{(i)}\|_0 = 1 \quad \text{for } i \in \{m+1, \ldots, m+n\}. \qquad (11)$$

By reducing the active set from 2 latents to 1, the split configuration ($\alpha = 1$) saves the sparsity budget. This creates a strong pressure on the optimization: the TopK constraint pushes the SAEs to use latents to represent mixed feature directions rather than maintaining atomic features, as this

saves capacity within the $k$-latent budget to reconstruct other features and lower the global reconstruction loss. $\quad\square$

### 4.2. Mitigating Splitting via Cross-Sample Consistency

Previous analysis shows that per-sample objectives ($\ell_1$ and TopK) cannot distinguish between atomic and split features. In contrast, the Minkowski inequality (Gruber, 1979) (Eq.4) for $p = 2$ offers a strict convexity condition to reverse this preference. Since the sum of norms for separate vectors is always greater than the norm of their sum, minimizing the sum of $\ell_2$ norms across the batch dimension, i.e. $\|Z_{:,1}\|_2 + \|Z_{:,2}\|_2$, encourages SAEs to consolidate shared semantic feature into a single latent.

We analyze this mechanism using the unified formulation in Table 2. We define a regularization term $\mathcal{L}_{\text{pair}}$ as the sum of the batch-norms for the two split latents:

$$\mathcal{L}_{\text{pair}}(\alpha) = \|Z_{:,1}\|_2 + \|Z_{:,2}\|_2$$
$$= \sqrt{\sum_{i=1}^{m}(z_1^{(i)})^2 + \sum_{i=m+1}^{m+n}((1-\alpha)z_1^{(i)})^2}$$
$$+ \sqrt{\sum_{i=m+1}^{m+n}(\alpha z_1^{(i)})^2 + \sum_{i=m+1}^{m+n}(z_2^{(i)})^2}. \tag{12}$$

To check if minimizing this term suppresses splitting, we calculate its gradient with respect to the splitting factor $\alpha$. The derivative $\frac{\partial \mathcal{L}_{\text{pair}}}{\partial \alpha}$ shows that the loss increases monotonically with $\alpha$ (meaning the penalty reduces splitting) as long as the following condition holds (derivation in Appendix A):

$$\alpha \geq \frac{1}{\sqrt{\frac{\sum_{i=1}^{m}(z_1^{(i)})^2}{\sum_{i=m+1}^{m+n}(z_2^{(i)})^2} + 1}}. \tag{13}$$

Empirical results from recent literature support this condition for practical SAE training. Leask et al. find that split latents retain high cosine similarity with their parent latents, implying the feature component magnitude far exceeds the residual ($\|z_1^{(i)}\| \gg \|z_2^{(i)}\|$). Additionally, Chanin et al. (2024) note large frequency gaps between parent and child features (e.g., $P(f_0) = 0.25$ vs $P(f_1) = 0.05$), which means the cumulative energy of the primary feature dominates the residual:

$$\sum_{i=1}^{m}(z_1^{(i)})^2 \gg \sum_{i=m+1}^{m+n}(z_2^{(i)})^2. \tag{14}$$

Under these settings, the right-hand side of Eq. 13 approaches zero. As a result, $\mathcal{L}_{\text{pair}}$ increases monotonically with $\alpha$ in the relevant domain. This confirms that a cross-sample $\ell_2$ penalty on redundant pairs theoretically ensures the consolidation of semantically related activations into a single consistent latent, mitigating both splitting and absorption.

We empirically verify the condition in Eq. 13 on our trained 65,536-latent SAEs using a 4M-token test set from SAEBench. As shown in Appendix K, the condition is satisfied in 88.1% of absorption pairs ($N = 4,555$), and the median ratio of the left-hand side to the right-hand side of Eq. 14 is 81.56. The remaining 11.9% of violated pairs correspond to marginal cases with low absorption coefficients, where neither the parent nor the child feature carries strong signal. This confirms that the theoretical guarantee holds in the vast majority of practically relevant cases, though we note its conditional nature.

## 5. Cross-Sample Consistency Regularization

Following the analysis in Section 4, we introduce $\mathbf{C^2R}$ (**C**ross-sample **C**onsistency **R**egularization). Minkowski inequality helps merge redundant features, but applying it to a large dictionary requires a careful approach. Here, we extend the pairwise analysis to the full SAE dictionary and examine the gradient dynamics that enable C²R to enforce both cross-sample consistency and latent orthogonality.

### 5.1. Generalizing from Pairwise to Multiple Latents

The derivation in Section 4 used a simple system with two latents for one parent feature and a child feature. However, standard SAEs have thousands of latents, and most represent distinct concepts. If we minimize the sum of norms across random pairs without selection, we would force independent features to merge, which causes feature collapse and reduces the SAEs' ability to resolve semantic differences.

We therefore need the regularization to be selective. It should only penalize latent pairs that look like split fragments or absorbed variations, while leaving independent features alone. Previous work on feature splitting (Chanin et al., 2024) and orthogonal constraints (Korznikov et al., 2025) shows that redundant latents usually have similar decoder weight directions. In contrast, distinct features tend to be nearly orthogonal in the high-dimensional space.

Based on this, we use the cosine similarity of decoder weights to detect redundancy. For each latent $i$, we find its nearest neighbor $j^*(i)$ in the decoder space:

$$j^*(i) = \arg\max_{j \neq i} \langle \hat{w}_i, \hat{w}_j \rangle, \quad \text{where } \hat{w} = \frac{w}{\|w\|_2}. \tag{15}$$

We then define the C²R loss by weighting the pairwise norm penalty with the squared rectified cosine similarity $\rho_{i,j^*(i)}^2 = \text{ReLU}(\langle \hat{w}_i, \hat{w}_{j^*(i)} \rangle)^2$:

$$\mathcal{L}_{\text{C}^2\text{R}}(X) = \frac{1}{k}\sum_{i=1}^{k} \rho_{i,j^*(i)}^2 \cdot \underbrace{(\|Z_{:,i}\|_2 + \|Z_{:,j^*(i)}\|_2)}_{S_{i,j^*(i)}}. \tag{16}$$

This weight acts as a gate. For distinct features where $\hat{w}_i \perp$

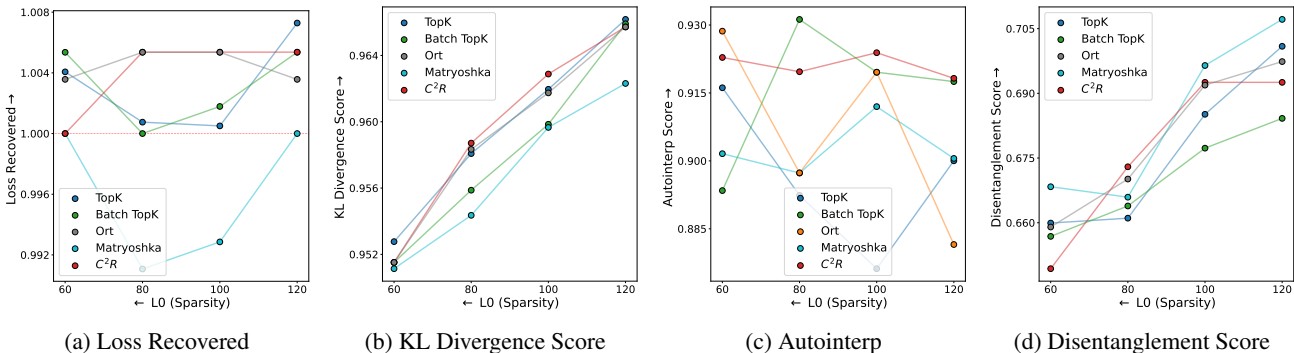

(a) Loss Recovered     (b) KL Divergence Score     (c) Autointerp     (d) Disentanglement Score

*Figure 2.* **Quantitative comparison of SAE performance across different sparsity levels.** (a) and (b) evaluate **reconstruction fidelity** using Cross-Entropy Loss and KL Divergence at the LLM output layer, respectively, (c) assesses SAE latent **interpretability** using the Autointerp score, and (d) measures the extent to which real-world features are disentangled into steerable latents. Notably, $C^2R$-enhanced Batch TopK SAE does not exhibit performance degradation compared to the vanilla Batch TopK baseline. Instead, it maintains competitive or superior results, particularly in KL Divergence and Autointerp scores, demonstrating that our proposed $C^2R$ constraint preserves these important capabilities.

$\hat{w}_j$, $\rho^2$ is zero, so the consistency constraint does not apply. For redundant features with high alignment, $\rho^2$ is large, which fully applies the regularization.

The final training objective adds this regularization to the standard SAE loss, which includes reconstruction and sparsity terms:

$$\mathcal{L}(X) = \mathcal{L}_{\text{SAE}}(X) + \lambda_{\text{C}^2\text{R}} \mathcal{L}_{\text{C}^2\text{R}}(X), \qquad (17)$$

where $\lambda_{\text{C}^2\text{R}}$ controls the weight of the cross-sample consistency term.

### 5.2. Gradient Analysis and Implicit Orthogonality

We can better understand $C^2R$ by looking at its gradient dynamics. The loss function is the product of a geometric alignment term ($\rho^2$) and an activation magnitude term ($S_{i,j^*(i)}$). The gradient with respect to the model parameters $\theta$ follows the product rule $\nabla(AB) = B\nabla A + A\nabla B$:

$$\nabla_\theta \mathcal{L}_{\text{C}^2\text{R}} \propto \underbrace{\rho^2 \cdot \nabla_\theta S_{i,j^*(i)}}_{\text{Consistency Gradient}} + \underbrace{S_{i,j^*(i)} \cdot \nabla_\theta(\rho^2)}_{\text{Orthogonality Gradient}}. \qquad (18)$$

This shows that $C^2R$ creates two simultaneous forces during optimization.

**1. Consistency Pressure** ($\rho^2 \nabla S$). The first term minimizes the sum of norms, scaled by the similarity weight $\rho^2$. As shown in Section 4, this uses the Minkowski inequality to push the splitting factor $\alpha$ toward 0. It merges the activation energy of redundant latents into a single one, which helps fix feature splitting and absorption.

**2. Implicit Orthogonality Pressure** ($S\nabla \rho^2$). The second term minimizes cosine similarity between decoder weights to encourage feature orthogonality, similar to the goals of OrtSAE (Korznikov et al., 2025). Unlike OrtSAE that typically applies a uniform penalty to all selected max-cosine

pairs, our method scales the penalty by $S_{i,j^*(i)}$ (the sum of feature activations). This allows $C^2R$ to adjust regularization based on feature frequency and magnitude. Strong, high-frequency features (large $S$) are subject to stricter orthogonality pressure to prevent redundancy. Conversely, for newly initialized or rare latents (small $S$), aggressive orthogonality enforcement can be counterproductive, potentially pushing them away from valid directions before they stabilize. By scaling the gradient with activation magnitude, $C^2R$ prevents such disruption, allowing developing latents to converge naturally. This mechanism promotes orthogonality while adapting the regularization strength to each feature's convergence state. We further clarify the implementation-level relationship between $C^2R$ and OrtSAE in Appendix L.

## 6. EXPERIMENTS

### 6.1. Baselines

Baselines include four different SAE architectures: TopK SAEs (Gao et al.), Batch TopK SAEs (Leask et al.), Matryoshka SAEs (Bussmann et al.), and OrtSAEs (Korznikov et al., 2025), as introduced in 2. For the main results in Figures 2 and 3, $C^2R$ uses Batch TopK as its base architecture to ensure a fair comparison with OrtSAE and Matryoshka SAE, both of which also build on Batch TopK. We evaluate performance by iterating over sparsity levels of $k \in \{60, 80, 100, 120\}$, ensuring a fair comparison between baselines and $C^2R$-enhanced Batch TopK SAE under equivalent sparsity conditions.. We run baselines and implement $C^2R$ on top of a public codebase (Karvonen, 2024), and the trained SAEs are evaluated with SAEBench (Karvonen et al.). Matryoshka SAEs use 5 layers, with the sizes of its 4 sub-SAEs set to $\{1/32, 1/8, 1/4, 1/2\}$ of the total latents.

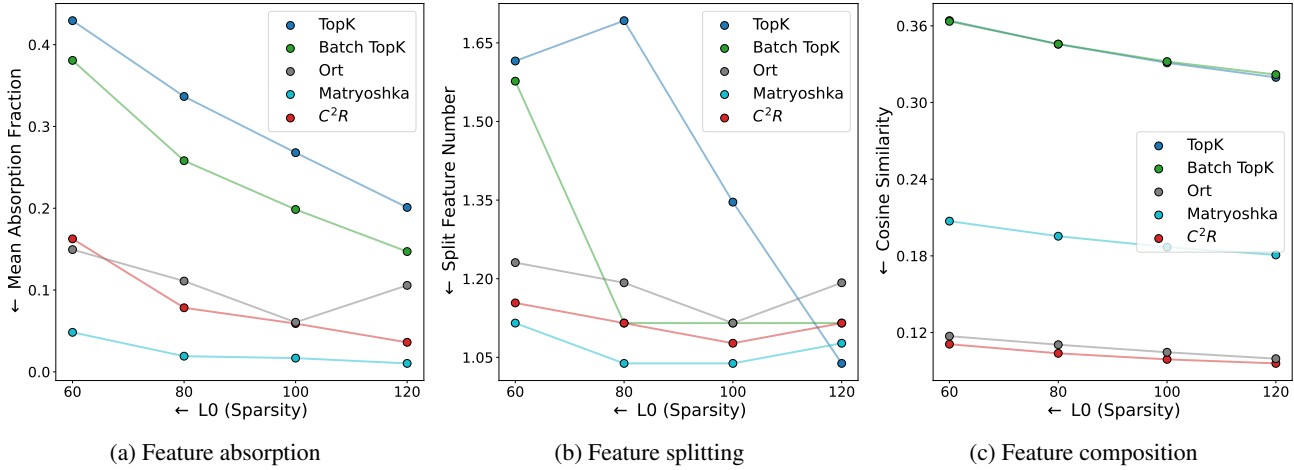

(a) Feature absorption       (b) Feature splitting       (c) Feature composition

*Figure 3.* **Analysis of feature structural metrics.** We compare feature absorption, splitting, and composition across different sparsity levels. Among methods that maintain high reconstruction fidelity (i.e., excluding Matryoshka SAEs), the proposed $C^2R$ constraint achieves the lowest rates of feature absorption and splitting. Furthermore, it achieves the optimal performance in feature decomposition, consistently yielding the lowest cosine similarity to ensure more atomic features.

### 6.2. Experiment Settings

We conduct systematic experiments on Gemma-2-2B (Team, 2024)[†] to validate the effectiveness of $C^2R$. Specifically, we sample a 500M-token subset from the OpenWebText dataset (Gokaslan et al., 2019)[‡] and use it to train a series of SAEs on the residual stream activations of the 12th layer of Gemma-2-2B. Each SAE has 65,536 latents, which is approximately 28 times the model's residual dimension, making it easier to observe feature absorption and feature splitting (Karvonen et al.). We performed a hyperparameter sweep for $\lambda_{C^2R}$ over the set $\{0.1, 0.5, 1, 5, 10\}$. We selected $\lambda_{C^2R} = 5$, as it represents the maximal regularization strength that does not degrade reconstruction fidelity. All SAEs are optimized with Adam using a learning rate of $2 \times 10^{-4}$, batch size of 2,048, and context length of 1,024.

**Computational Efficiency.** Computing the pairwise cosine similarity for a large dictionary (e.g., $k = 65,536$) imposes a quadratic $O(k^2)$ computational complexity and substantial memory overhead. To ensure computational feasibility and maintain a fair comparison with the state-of-the-art baseline, we adopt the efficient optimization strategy used in OrtSAEs (Korznikov et al., 2025). Specifically, we employ a block-wise computation strategy with a chunk size of 8,192 and compute the consistency regularization term every 5 training steps, scaling the coefficient $\lambda_{C^2R}$ accordingly. This approach reduces the overhead to negligible levels while preserving performance (detailed in Appendix B).

---

[†]Gemma Terms of Use
[‡]Creative Commons Zero v1.0 Universal

### 6.3. Metrics

To holistically assess the relative performance of SAEs when integrating $C^2R$, our evaluation utilizes a comprehensive set of seven key metrics. These metrics, implemented using the code framework from SAEBench (Karvonen et al.), cover four areas: **reconstruction fidelity**, **feature hierarchy**, **interpretability**, and **disentanglement**. Specifically, our evaluation consists of six key metrics: **Loss Recovered**, **KL Div. Score**, **AutoInterp**, **Split Num**, **Absorption Rate**, **Composition**, and **Disentanglement**. Detailed description of these metrics is in E.

### 6.4. Reconstruction Fidelity

As shown in Figure 2, integrating $C^2R$ preserves reconstruction fidelity of the backbone SAEs. Figure 2a reports the Loss Recovery metric (details in Appendix E), where a value $\geq 1$ indicates that the SAE-reconstructed LLM activations yield a cross-entropy loss lower than or equal to the original LLM. We observe that across the tested sparsity levels, only Matryoshka SAEs exhibit performance loss, while other methods maintain full recovery. Figure 2b displays the KL Divergence Score, which measures the shift in the LLM's output logit distribution. Higher scores correspond to smaller deviations from the original distribution. The results demonstrate that SAEs trained with the $C^2R$ constraint maintain high reconstruction fidelity, strictly adhering to the original model's behavior.

### 6.5. Interpretability

Following the evaluation framework of (Paulo et al.), we assess the interpretability of all SAEs using the *AutoInterp* metric. The results in Figure 2c show that adding $C^2R$ does

not significantly affect interpretability. For each SAE, we sampled 128 latents and constructed prompts based on their activations over a 2M-token input. We prompt an LLM explainer to generate concise and comprehensive textual descriptions for each latent from 15 observed samples, and prompt an independent LLM judge predicted whether each latent would activate on 15 unseen test samples. The detailed prompts used for both the LLM explainer and LLM judger, as well as the instructions provided to human annotators, are included in Appendix C.

We employ GPT-5-mini (OpenAI, 2025) as the LLM judge and validate its reliability via a user study. GPT-5-mini achieved a 97.3% match rate with humans and a Pearson $r$ of 0.74, confirming its suitability as a proxy evaluator for automated interpretability assessment. Details are in Appendix D. To further verify that the results are not sensitive to the choice of judge model, we re-evaluate all AutoInterp scores using GPT-5. The per-method Pearson correlations between GPT-5-mini and GPT-5 scores range from 0.81 to 0.94 (all $p < 0.001$), confirming that the interpretability rankings are consistent across judge models. Full results are in Table 7.

### 6.6. Disentanglement

We evaluate disentanglement using the RAVEL benchmark (Huang et al., 2024), which employs interchange interventions to determine if specific SAE latents causally control individual attributes such as continent or gender. The metric averages cause and isolation scores to measure how well the SAE governs a target concept without affecting unrelated ones. As illustrated in Figure 2d, Batch TopK SAEs trained with the C²R constraint achieve performance comparable to the vanilla baseline. This demonstrates that our regularization effectively preserves a disentangled and manipulable latent space.

### 6.7. Feature Splitting and Feature Absorption

Figures 3a and 3b illustrate the evaluation of feature absorption and splitting. Our proposed C²R constraint effectively mitigates both issues compared to the vanilla Batch TopK baseline and outperforms the orthogonal constraint in OrtSAE. Although Matryoshka SAEs achieve lower absorption and splitting scores, this improvement comes at the expense of reconstruction fidelity, as evidenced in Figure 2a and 2b. Therefore, among methods that maintain the original model's performance, the C²R constraint strikes the best balance, achieving the lowest levels of absorption and splitting while preserving latent quality and reconstruction capabilities.

### 6.8. Feature Composition

Following Bussmann et al., we quantify feature composition by measuring the average maximum cosine similarity between latent vectors. High similarity implies that multiple latents represent overlapping information, indicating a lack of atomicity. As shown in Figure 3c, our approach achieves the optimal performance on this metric. The C²R constraint yields significantly lower cosine similarity compared to vanilla Batch TopK and Matryoshka SAEs, and marginally outperforms OrtSAE. This result confirms that our method effectively minimizes redundancy, producing a set of highly distinct and atomic features.

### 6.9. Robustness and Generalization

We conduct a series of additional experiments to verify the robustness and generalization of C²R across random seeds, model scales, layers, and training data. Detailed results are provided in Appendix G.

**Statistical validation.** To assess the stability of our main results, we perform 5 independent training runs with different random seeds at sparsity level $k=100$ on Gemma-2-2B layer 12. As shown in Table 8, C²R consistently achieves the best feature composition and absorption with narrow confidence intervals, confirming that the reported improvements are statistically robust.

**Scaling to larger models.** We evaluate C²R on two current-generation models, Qwen3-8B and Llama-3-8B, both at layer 20 with sparsity level $k=100$. Tables 9 and 10 show that C²R maintains its effectiveness at the 8B scale: it achieves the lowest composition and absorption while preserving reconstruction fidelity, demonstrating that our method generalizes beyond the Gemma-2-2B setting.

**Cross-layer consistency.** Beyond the layer 12 experiments in the main evaluation, we additionally train SAEs on Gemma-2-2B layer 20 (Table 11). C²R continues to achieve optimal or near-optimal structural metrics at a deeper layer, confirming cross-layer consistency.

**Sensitivity to training data.** We investigate the effect of training data scale and composition by (1) extending the training corpus from 500M to 1B tokens on OpenWebText, and (2) training on The Pile (Gao et al., 2020), a structurally diverse dataset spanning academic, internet, prose, and dialogue domains. As shown in Tables 12 and 13, C²R consistently achieves optimal or near-optimal structural feature metrics in both settings, confirming that our method is not sensitive to the scale or composition of the training data.

### 6.10. Ablation Study

We ablate the key design choices of C²R and study the sensitivity to the regularization strength $\lambda_{C^2R}$. All experiments

are conducted on Gemma-2-2B layer 12 at sparsity level $k=100$. Detailed results are provided in Appendix H.

**Component ablation.** We evaluate two variants: (1) **NoNNR**, which removes the nearest-neighbor restriction and applies the consistency penalty to all latent pairs; and (2) **NoRCG**, which removes the ReLU cosine gate and applies the penalty without directional selectivity. As shown in Table 14, removing either component imposes an overly aggressive merging constraint. While this severely penalizes feature absorption, it does so at the unacceptable cost of significantly degrading reconstruction fidelity and inflating composition scores. This demonstrates that the cautious, gated constraints of $C^2R$ are necessary to mitigate feature pathologies without destroying dictionary utility.

**Hyperparameter sensitivity.** We sweep $\lambda_{C^2R} \in \{0.1, 0.5, 1, 5, 10\}$ to understand its effect on the trade-off between reconstruction and structural metrics (Table 15). As $\lambda_{C^2R}$ increases, composition and absorption decrease monotonically, while reconstruction fidelity remains stable up to $\lambda_{C^2R}=5$ and begins to degrade at $\lambda_{C^2R}=10$. We select $\lambda_{C^2R}=5$ as the default, as it maximizes the reduction in feature composition and absorption without sacrificing reconstruction quality.

# 7. Compatibility with Other SAE Architectures

A key design goal of $C^2R$ is to serve as a general-purpose regularizer that can be applied on top of any SAE architecture. To verify this, we evaluate $C^2R$ when integrated with different backbone architectures beyond the Batch TopK used in our main experiments. All experiments are conducted on Gemma-2-2B layer 12 at sparsity level $k=100$. Detailed results are provided in Appendix I.

**Integration with TopK and OrtSAE.** We apply $C^2R$ to both the standard TopK and OrtSAE backbones. As shown in Table 16, $C^2R$ consistently and substantially improves feature structural metrics regardless of the base architecture, without degrading reconstruction fidelity. Notably, TopK + $C^2R$ achieves a significant reduction in Composition and Absorption, demonstrating that $C^2R$'s benefits are not tied to the Batch TopK backbone but arise from the cross-sample consistency mechanism itself.

**Integration with AbsTopK.** To evaluate whether $C^2R$ adapts to bi-directional encoder architectures, we apply it on top of AbsTopK, which uses signed activations. As shown in Table 16, $C^2R$ effectively improves feature hierarchy in this signed setting while preserving reconstruction fidelity. The ReLU cosine gate remains appropriate because feature absorption and splitting imply cosine similarity $> 0$ between the involved decoder directions. By targeting only positive similarities, $C^2R$ avoids collateral damage to non-

pathological negative correlations, ensuring training stability in signed encoder settings.

## 7.1. Downstream Causal Intervention Tasks

We further evaluate the practical utility of learned SAE features on two downstream causal intervention tasks from SAEBench. Spurious Correlation Removal (SCR) measures the ability to remove spurious correlations by ablating relevant SAE features, and Targeted Probe Perturbation (TPP) measures the precision of causal interventions via targeted feature perturbation. As shown in Table 17 in Appendix J, $C^2R$ outperforms TopK, Batch TopK, and Ort on both tasks and achieves comparable performance to Matryoshka, indicating that the improved feature structure transfers to downstream causal applications.

# 8. Conclusion

In this work, we introduced $C^2R$, a theoretically grounded constraint aimed at improving feature hierarchy in sparse autoencoders. By leveraging the Minkowski inequality, our approach provides a rigorous guarantee against feature splitting and absorption. Extensive empirical evaluations show that incorporating the $C^2R$ constraint significantly enhances the structural quality of the learned dictionary, reducing feature composition and minimizing redundancy. Importantly, our approach preserves reconstruction fidelity, latent interpretability, and disentanglement capabilities on par with the baselines. These results highlight the effectiveness of the cross-sample consistency regularization in addressing the trade-off between feature atomicity and model fidelity, offering a robust solution for large language model analysis.

# 9. Limitations

The theoretical guarantee in Eq. 13 is conditional and holds in 88.1% of absorption pairs in our verification (Appendix K). A fully unconditional guarantee remains an open problem. Although we extend our evaluation to Qwen3-8B and Llama-3-8B, we have not verified scalability to larger models or diverse architectures such as Mixture-of-Experts. Our AutoInterp evaluation uses the proprietary GPT-5-mini as the judge model. We verify high consistency with GPT-5 and human evaluations, but this still introduces a reproducibility constraint. In addition, $C^2R$ may suppress legitimate polysemanticity when distinct features have moderately aligned decoder directions. We find that $C^2R$ maintains over 99% dictionary utilization (Appendix N), but the interaction with polysemantic features needs further study.

## Acknowledgements

This work was supported by the National Natural Science Foundation of China (NSFC) (No. U24A20253, NO. 62476279, NO. 92470205, NO. U2436209), Scientific Research Innovation Capability Support Project for Young Faculty, Major Innovation & Planning Interdisciplinary Platform for the "Double-First Class" Initiative, Renmin University of China, the Fundamental Research Funds for the Central Universities, and the Research Funds of Renmin University of China No. 24XNKJ18. Supported by fund for building world-class universities (disciplines) of Renmin University of China and Public Computing Cloud, Renmin University of China.

## Impact Statement

This work advances the methodology of sparse dictionary learning for neural networks. By improving the fidelity of feature extraction, we aim to enable a more granular understanding of large language model internals. Such interpretability is essential for auditing model behavior, identifying latent failure modes, and verifying safety properties prior to deployment. However, we acknowledge that deeper insights into model representations can also be leveraged to improve model efficiency or steerability, potentially accelerating the development of powerful systems and amplifying the societal risks associated with their deployment.

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

# A. C²R Promotes Cross-sample Consistency

When feature absorption occurs, the C²R Loss is:

$$\mathcal{L}_{\text{C}^2\text{R}}(X) = \sqrt{\sum_{i=1}^{m}(z_1^{(i)})^2 + \sum_{i=m+1}^{m+n}((1-\alpha)z_1^{(i)})^2}$$

$$+ \sqrt{\sum_{i=m+1}^{m+n}(\alpha z_1^{(i)})^2 + \sum_{i=m+1}^{m+n}(z_2^{(i)})^2}. \tag{19}$$

Taking the partial derivative with respect to $\alpha$ yields:

$$\frac{\partial \mathcal{L}_{\text{C}^2\text{R}}(X)}{\partial \alpha} = \frac{(\alpha-1)\sum_{i=m+1}^{m+n}(z_1^{(i)})^2}{\sqrt{\sum_{i=1}^{m}(z_1^{(i)})^2 + \sum_{i=m+1}^{m+n}((1-\alpha)z_1^{(i)})^2}}$$

$$+ \frac{\alpha \sum_{i=m+1}^{m+n}(z_1^{(i)})^2}{\sqrt{\sum_{i=m+1}^{m+n}(\alpha z_1^{(i)})^2 + \sum_{i=m+1}^{m+n}(z_2^{(i)})^2}}. \tag{20}$$

The correct loss should be proportional to $\alpha \in [0, 1]$:

$$\frac{\partial \mathcal{L}_{\text{C}^2\text{R}}(X)}{\partial \alpha} \geq 0. \tag{21}$$

Introducing the shorthand notation:

$$A = \sum_{i=1}^{m}(z_1^{(i)})^2$$

$$B = \sum_{i=m+1}^{m+n}(z_2^{(i)})^2 \tag{22}$$

$$C = \sum_{i=m+1}^{m+n}(z_1^{(i)})^2$$

The inequality simplifies to:

$$\frac{\alpha C}{\sqrt{\alpha^2 C + B}} \geq \frac{(1-\alpha)C}{\sqrt{A + (1-\alpha)^2 C}}. \tag{23}$$

The final form is:

$$\alpha \geq \frac{1}{\sqrt{\frac{\sum_{i=1}^{m}(z_1^{(i)})^2}{\sum_{i=m+1}^{m+n}(z_2^{(i)})^2} + 1}} \tag{24}$$

# B. Efficient Implementation Details

Applying the C²R constraint naively requires computing the cosine similarity between all pairs of decoder weight vectors, resulting in a $k \times k$ similarity matrix. For a dictionary size of $k = 65,536$, this operation requires $O(k^2)$ memory and computation, which significantly slows down training and increases VRAM usage.

To address this, we implement two engineering optimizations following the methodology of OrtSAE (Korznikov et al., 2025):

**Chunk-wise Approximation.** Instead of searching for the nearest neighbor $j^*(i)$ across the entire dictionary, we randomly permute the feature indices at each step and partition the dictionary into smaller blocks (chunks). For a dictionary size $k$ and chunk size $C$, we partition the latents into $N = k/C$ chunks. The nearest neighbor search and loss computation are then restricted to within each chunk.

$$j^*(i) \approx \underset{j \in \text{Chunk}(i), j \neq i}{\arg\max} \langle \hat{w}_i, \hat{w}_j \rangle. \tag{25}$$

In our experiments, we use a chunk size of $C = 8,192$. This reduces the complexity from $O(k^2)$ to $O(C \cdot C)$. Since high-cosine similarity features (redundant pairs) are rare and the permutation is randomized at every step, the probability of a redundant pair falling into the same chunk accumulates rapidly over training steps, ensuring the regularization remains effective.

**Periodic Updates.** To further reduce the computational overhead, we compute the $C^2R$ loss and its gradients only every $T$ training steps rather than at every iteration. To maintain the same effective regularization strength over time, we scale the regularization coefficient $\lambda_{C^2R}$ by the period $T$:

$$\mathcal{L}_{\text{step } t}(X) = \mathcal{L}_{\text{SAE}}(X) + (\mathbb{1}_{t \bmod T=0} \cdot T \cdot \lambda_{C^2R})\mathcal{L}_{C^2R}(X). \tag{26}$$

We set $T = 5$ for all experiments of OrtSAEs and our approach. This configuration aligns our training cost with other SAEs' training times, adding only negligible overhead (slightly more than the findings in OrtSAE, where overhead was reduced to $< 10\%$).

## C. Prompts for LLMs and Instructions for Human Annotators

This appendix presents the detailed prompts used in the interpretability evaluation. The prompts were designed to elicit consistent reasoning from both LLM-based and human evaluators. Two types of LLMs were employed: an *Explainer* to describe latent semantics and a *Predictor* (or Judge) to estimate latent activation likelihood. For comparison, human annotators followed analogous instructions.

### C.1. LLM Explainer Prompt

An example of the prompts used to generate textual descriptions for each latent activation is shown in Table 3.

### C.2. LLM Predictor Prompt

An example of the prompts used by the LLM judge (the predictor) to decide whether the described latent would activate for each unseen test sample is shown in Table 4.

### C.3. Human Annotator Instruction

Human annotators were provided with the latent's same activating samples and unseen samples as the LLM explainer and the LLM predictor. An example of the instructions is shown in Table 5.

## D. User Study Details

We recruited three human annotators with high-school-level English proficiency, who replicated the explainer–judge process on 30 latents sampled from Batch TopK SAE and its $C^2R$-enhanced variant at layer 12 of *Gemma-2-2B*, covering five $L_0$ settings. Each annotator produced 450 activation predictions. The probability of agreement between human annotators and GPT-5-mini, as well as the Pearson correlation coefficient (Pearson, 1895) between human- and LLM-derived scores, are summarized in Table 6. GPT-5-mini achieved a **97.3%** match rate with humans and a Pearson $r$ of **0.74**, confirming its suitability as a proxy evaluator for automated interpretability assessment.

The annotators are recruited from the university, and the compensation was set according to the standard payment guidelines for on-campus research participation.

To verify that our results are not sensitive to the specific judge model, we re-evaluate the AutoInterp scores of all methods using GPT-5 and compute the per-method Pearson correlation with the original GPT-5-mini scores. As shown in Table 7, all correlations exceed 0.81 with $p < 0.001$, confirming high consistency.

## E. Detailed Metrics Description

The following six key metrics are used to evaluate the performance of SAEs integrated with $C^2R$:

- **Loss Recovered** This metric is the primary measure of **reconstruction fidelity**. It quantifies the degree to which an

| **Prompt example for the LLM explainer** |
| --- |
| We're studying neurons in a neural network. Each neuron activates on some particular word/words/substring/concept in a short document. The activating words in each document are indicated with «token[act:activation]».

We will give you a list of ACTIVATE documents, where the neuron fires, ordered by strength. Look at the marked parts of the ACTIVATE documents and summarize in a single sentence what the neuron is activating on. Try not to be overly specific or overly broad. Your explanation should cover most or all activating words. Pay attention to things like capitalization and punctuation if relevant. Keep the explanation as short and simple as possible, limited to 30 words or less. Omit punctuation and formatting. Some examples: "This neuron activates on the word 'knows' in rhetorical questions", and "This neuron activates on verbs related to decision-making and preferences", and "This neuron activates on the substring 'Ent' at the start of words", and "This neuron activates on text about government economic policy".

The relevant documents are given below:

ACTIVATE (1). see he was enjoying the other shapes too – the« round[act:55.5]» bowl and basket and the books underneath them, the
ACTIVATE (2). The tube may be« cylindrical[act:16.75]» (or conical) with« circular[act:55.0]», rectangular or any desired cross section.↩By
ACTIVATE (3). the factors affecting the appearance of impact craters↩The« circular[act:53.0]» features so obvious on the Moon's surface are
ACTIVATE (4). example, the simple cylindrical case which cylinder has a« circular[act:52.5]» cross section, will be considered in detail. If
ACTIVATE (5). when Picasaweb closed. They consist of a« circular[act:50.0]» emitter Psurrounded by a ring shaped base N
ACTIVATE (6). home.↩A mosquito bite appears as an itchy« round[act:50.0]», red, or pink skin bump. It'
ACTIVATE (7). of the large square sew-on. Take the« round[act:48.25]» sew-on and glue it on the left side
ACTIVATE (8). multiple locations to accommodate various connection sizes and elevations.« Round[act:47.75]» or rectangular shapes available per design specifications. Knock-
ACTIVATE (9). . Tie and suspend with gold thread from either our« round[act:44.25]» hoop or a stick of your choice.↩Hang
ACTIVATE (10). bles has been played over the centuries with everything from« rounded[act:41.75]» sea pebbles to fruit pits, today the game is
ACTIVATE (11). discount on Flashflight.com's most popular« circular[act:41.25]» and spherical objects. From now until March 1
ACTIVATE (12). front extension, stone corbelling under eaves,« circular[act:41.25]» light in gable peak, slender turret with Christian cross
ACTIVATE (13). piles of rock (called ejecta) around the« circular[act:41.25]» hole as well as↩bright streaks of target material
ACTIVATE (14). waterproof back and an outer back with 16« round[act:41.0]» openings. Manufactured in 1967,
ACTIVATE (15). A showcases her gorgeous slender body with swollen breasts,« round[act:40.75]» butt, and slender toes on the veranda.↩ |

*Table 3.* Prompt example for LLM explainer to explain a latent based on its activations.

---

**Prompt example for the LLM predictor**

We're studying neurons in a neural network. Each neuron activates on some particular word/words/substring/concept in a short document. You will be given a short explanation of what this neuron activates for, and then be shown 15 example sequences in random order. You will have to return a comma-separated list of the examples where you think the neuron should activate at least once, on ANY of the words or substrings in the document. For example, your response might look like "1, 2, 6, 9, 12". Try not to be overly specific in your interpretation of the explanation. If you think there are no examples where the neuron will activate, you should just respond with "None". You should include nothing else in your response other than comma-separated numbers or the word "None" - this is important.

Here is the explanation: this neuron fires on words describing round or circular shapes including round circular rounded and cylindrical.

Here are the examples:

1. in South Africa • Uganda) · Asia (in China • India • Myanmar • Pakistan • Taiwan • Japan
2. ized was either beheaded or shot at point blank range." more »←A Syrian mother and widow was tortured
3. method' anyone can do that but getting the right mindset to succeed. This is something most traders simply cannot
4. Facebook Be Fixed?←CMS Wire (May 24, 2012) - Facebook
5. . Finished in a weathered brown and accented with a circular polished silver bezel. The metal dial has polished silver
6. when a customer has changed his or her mind about a transaction, or when an error has occurred, the
7. . Several of you have reached out to us and to our colleagues across the Administration. You've warned
8. crater is that←you cannot see it. Its circular structure is nearly a kilometer below the←surface and
9. sound crazy? Okay?←DW:I'm just going to tell you the truth.←THE
10. am I missing some key information here?<eos>The eleventh round of 2020 Monster Energy Super
11. awesome Flashflight Light-Up Flying Discs are circular, and our equally saucy Meteorlight LED Light
12. rather than doubling Defense on Dodge←Strength ••, Brawl •←Add Brawl rather than doubling Defense on Dodge
13. achieve a perfectly snug fit. Lastly the grain is circular-grained, after which the stone will not move
14. . Instead he drew a Dalek with two big round holes in it, and a guy catching a baseball
15. abb, Sean McDermott, Kevin Kolb). If the Eagles are waiting for a Packers assistant, the best

*Table 4.* Prompt example for LLM judger to predict latent activations based on its explanation generated by the LLM explainer.

**Instruction example for human annotators**

We're studying neurons in a neural network. Each neuron activates on some particular word/words/substring/concept in a short document. The activating words in each document are highlighted.

We will give you a list of ACTIVATE documents (where the neuron fires, ordered by strength), please look at the marked parts of the ACTIVATE documents. Summarize in a single sentence what the neuron is activating on. Try not to be overly specific or overly broad. Your explanation should cover all activating words. Pay attention to things like capitalization and punctuation if relevant. Keep the explanation as short and simple as possible, limited to 30 words or less. Omit punctuation and formatting.

Some examples: "This neuron activates on the word 'knows' in rhetorical questions", and "This neuron activates on verbs related to decision-making and preferences", and "This neuron activates on the substring 'Ent' at the start of words", and "This neuron activates on text about government economic policy".

The relevant documents are given below:

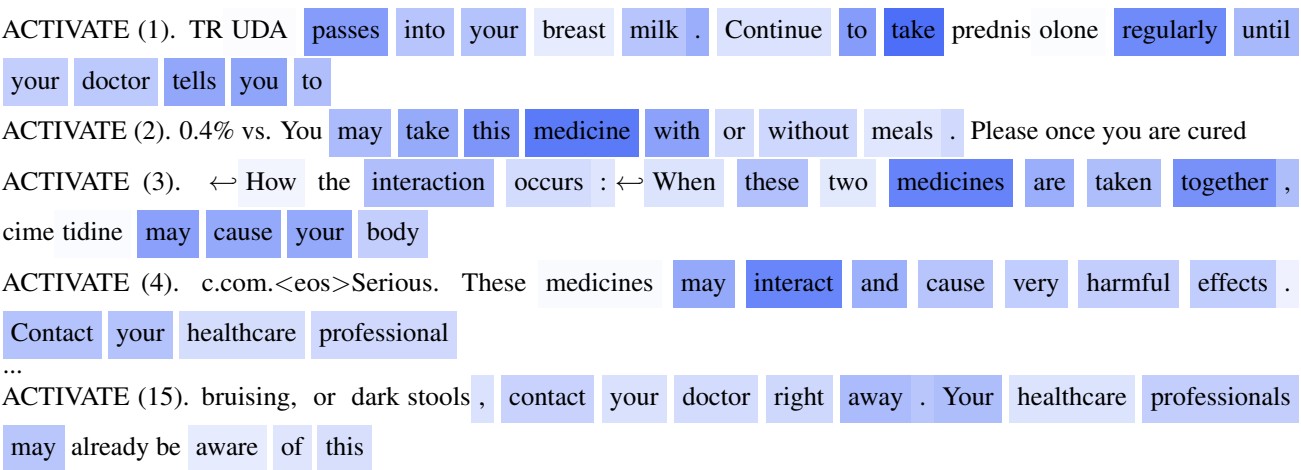

Based on your explanation of what this neuron activates for, please review the following 15 examples and indicate if you believe the neuron should activate at least once on ANY of the words or substrings within the document. Provide the corresponding text IDs. For instance, your response might look like "2, 3, 5, 6, 13". Avoid being overly specific in your interpretation of the explanation.

Here are the examples:

1. off-the-wall in this first directorial effort from the 49-year-old Belgian
2. your organization.<eos>The Academy of Motion Picture Arts and Sciences which is best known for organizing the Oscars has
3. Realtors and the Mortgage Bankers Association.↩But this time, lobbyists are worried. That's because
4. considered a natural antihistamine. Valtrex is used to treat herpes zoster and herpes simplex and,

...

15. of the stomach and intestines.↩Be sure to tell your doctor if you experience any of these side effects

*Table 5.* Instruction example for human annotators to predict latent activation.

|            | Prediction Match | Pearson $r$ |
|------------|------------------|-------------|
| Annotator 1 | 97.8%            | 0.74        |
| Annotator 2 | 96.7%            | 0.63        |
| Annotator 3 | 97.6%            | 0.86        |
| Average     | 97.3%            | 0.74        |

*Table 6.* User study comparing GPT-5-mini (OpenAI, 2025) and human annotators in the automated interpretability task. The table reports the match rate and Pearson correlation (Pearson, 1895) between human- and LLM-derived AutoInterp scores.

| Method | GPT-5-mini | GPT-5 | Pearson $r$ |
|--------|-----------|-------|-------------|
| $C^2R$ | 0.9239    | 0.9218 | 0.94       |
| Batch TopK | 0.9208 | 0.9161 | 0.89       |
| Matryoshka | 0.9120 | 0.9031 | 0.89       |
| Ort    | 0.9196    | 0.9153 | 0.81       |
| TopK   | 0.8762    | 0.8735 | 0.94       |

*Table 7.* Consistency between GPT-5-mini and GPT-5 as AutoInterp judges. All Pearson correlations are significant ($p < 0.001$).

SAE can preserve the original language model's Next-Token Prediction performance after its internal activations are reconstructed. It is defined as: Loss Recovered $= \frac{(H^* - H_0)}{(H_{orig} - H_0)}$ where $H_{orig}$ is the original cross-entropy loss, $H^*$ is the loss after replacement with SAE-reconstructed activations, and $H_0$ is the loss after zero-ablating the original activations. A higher value indicates better reconstruction fidelity.

- **KL Div. Score (Gao et al.)** As a complementary measure of reconstruction quality, this metric assesses how effectively the SAE's reconstruction recovers the model's output distribution from a zero-ablated baseline. It is a normalized score that quantifies the reduction in Kullback-Leibler (KL) divergence between the output logits and the original model's logits distribution ($P_{orig}$). The score is calculated as: KL Div. Score $= 1 - \frac{D_{KL}(P_{SAE} \| P_{orig})}{D_{KL}(P_{ablated} \| P_{orig})}$ where $D_{KL}(P_{ablated} \| P_{orig})$ is the KL divergence when the activation is zero-ablated, and $D_{KL}(P_{SAE} \| P_{orig})$ is the KL divergence when the activation is replaced by the SAE reconstruction. This score is bounded between 0 and 1, where a higher score signifies superior reconstruction performance relative to the zero-ablated state.

- **AutoInterp (Paulo et al.)** This metric evaluates the human-understandability of learned latents using LLMs. It operates in two stages: an LLM generates a feature description based on activating inputs, and another LLM judge uses this description to predict latent activation on new sequences. The prediction accuracy serves as the AutoInterp score.

- **Split Num (Chanin et al., 2024)** A diagnostic metric focusing on feature splitting. This metric serves as a proxy for the granularity and non-redundancy of the learned latents. It is measured by identifying a single high-level concept (e.g., all tokens starting with a specific letter) and counting the minimum number of distinct SAE latents required to significantly improve classification performance on a probe for that concept. A higher count can indicate that more latent are fragmented into distinct components.

- **Absorption (Chanin et al., 2024)** This metric focuses on feature absorption. It measures the tendency of an SAE to learn two coupled hierarchical features (e.g., A and "B excluding A") instead of two independent features (A and B). The metric is calculated by diagnosing the activation patterns of the general SAE latents: measuring the frequency at which they fail to activate when a token-aligned child latent is present in the input. A lower score is desirable, indicating better feature isolation and reduced feature absorption.

- **Max CosSim** This metric quantifies the maximum cosine similarity between the decoder weight vectors of all learned latents, reflecting the feature composition level of SAEs (Bussmann et al.). High similarity suggests significant directional overlap or redundancy among SAE latents.

- **Disentanglement (Huang et al., 2024)** This benchmark evaluates the ability of interpretability methods to disentangle independent attributes within language model representations. It utilizes interchange interventions to test whether a targeted feature (e.g., city country) can be modified without affecting other related attributes (e.g., city language). The performance is summarized by the disentangle score: Disentangle Score $= \frac{1}{2}$(Cause + Iso) where **Cause** measures the

success rate of changing the target attribute's value through intervention, and **Iso** (Isolation) measures the frequency with which non-target attributes remain unchanged. A high score indicates that the SAE has successfully localized individual concepts into independent, causal units.

## F. GPU Budget

We ran all SAE training experiments utilizing an NVIDIA H800 GPU, consuming a total of 300 GPU hours and achieving a peak memory utilization of 65 GB.

Evaluating a trained SAE across all metrics required approximately 1 GPU hour. The breakdown of evaluation time per SAE is as follows:

- Reconstruction Fidelity Metrics: 30 minutes

- Interpretability Analysis: 10 minutes

- Feature Hierarchy Metrics: 10 minutes

- Disentanglement Metrics: 15 minutes

In terms of memory footprint, the reconstruction fidelity evaluation was the most demanding, requiring up to 50 GB of VRAM. The interpretability and feature hierarchy analyses were less memory-intensive, each requiring approximately 15 GB.

## G. Robustness and Generalization

| Method | KL Score ↑ | Interp ↑ | Composition ↓ | Absorption ↓ | Split ↓ |
|---|---|---|---|---|---|
| Batch TopK | $\underline{0.9621 \pm 0.0006}$ | $\mathbf{0.9311 \pm 0.0098}$ | $0.3325 \pm 0.0003$ | $0.1908 \pm 0.0240$ | $1.0962 \pm 0.0722$ |
| Matryoshka | $0.9597 \pm 0.0008$ | $\mathbf{0.9311 \pm 0.0120}$ | $0.1870 \pm 0.0003$ | $\mathbf{0.0360 \pm 0.0066}$ | $1.0673 \pm 0.0361$ |
| Ort | $0.9615 \pm 0.0006$ | $0.9297 \pm 0.0082$ | $\underline{0.1038 \pm 0.0004}$ | $0.0819 \pm 0.0203$ | $\underline{1.0577 \pm 0.0218}$ |
| C²R | $\mathbf{0.9628 \pm 0.0004}$ | $\underline{0.9309 \pm 0.0052}$ | $\mathbf{0.0991 \pm 0.0003}$ | $\underline{0.0656 \pm 0.0185}$ | $\mathbf{1.0385 \pm 0.0238}$ |

*Table 8.* Statistical validation with 95% confidence intervals over 5 independent runs (Gemma-2-2B, layer 12, $k$=100). **Bold** and underline indicate the best and second-best values, respectively.

| Method | KL Score ↑ | Interp ↑ | Composition ↓ | Absorption ↓ | Split ↓ |
|---|---|---|---|---|---|
| Batch TopK | **0.9889** | 0.9220 | 0.2677 | 0.0074 | 1.1154 |
| Matryoshka | 0.9870 | 0.9277 | 0.1422 | 0.0032 | 1.1154 |
| Ort | 0.9886 | 0.9214 | 0.0945 | 0.0034 | 1.1154 |
| C²R | 0.9874 | **0.9333** | **0.0675** | **0.0018** | 1.1154 |

*Table 9.* Results on Qwen3-8B (layer 20, $k$=100).

| Method | KL Score ↑ | Interp ↑ | Composition ↓ | Absorption ↓ | Split ↓ |
|---|---|---|---|---|---|
| Batch TopK | **0.9860** | 0.9405 | 0.3334 | 0.1470 | 1.1923 |
| Matryoshka | 0.9853 | **0.9552** | 0.1803 | **0.0349** | 1.1154 |
| Ort | 0.9859 | 0.9265 | 0.1200 | 0.0979 | **1.0385** |
| C²R | 0.9858 | 0.9443 | **0.0816** | 0.0410 | **1.0385** |

*Table 10.* Results on Llama-3-8B (layer 20, $k$=100).

| Method | KL Score ↑ | Interp ↑ | Composition ↓ | Absorption ↓ | Split ↓ |
|--------|-----------|----------|---------------|--------------|---------|
| TopK | 0.9720 | 0.9307 | 0.3270 | 0.1325 | 1.5769 |
| Batch TopK | 0.9706 | 0.9323 | 0.3329 | 0.1087 | 1.3462 |
| Matryoshka | **0.9727** | 0.9410 | 0.1781 | **0.0205** | **1.2308** |
| Ort | 0.9722 | 0.9543 | 0.1000 | 0.0352 | 1.2692 |
| $C^2R$ | **0.9727** | **0.9570** | **0.0781** | 0.0260 | 1.2692 |

*Table 11.* Results on Gemma-2-2B layer 20 ($k$=100).

| Method | KL Score ↑ | Interp ↑ | Composition ↓ | Absorption ↓ | Split ↓ |
|--------|-----------|----------|---------------|--------------|---------|
| TopK | **0.9631** | 0.9196 | 0.3416 | 0.2987 | 1.2692 |
| Batch TopK | 0.9621 | **0.9313** | 0.3430 | 0.2149 | 1.1538 |
| Matryoshka | 0.9598 | 0.8870 | 0.1892 | **0.0241** | 1.0769 |
| Ort | 0.9629 | 0.9072 | 0.1053 | 0.1058 | **1.0385** |
| $C^2R$ | **0.9631** | 0.9302 | **0.0870** | 0.0654 | **1.0385** |

*Table 12.* Results with 1B training tokens on OpenWebText (Gemma-2-2B, layer 12, $k$=100).

## H. Ablation Study

Table 14 reports the component ablation results, where removing the nearest-neighbor restriction (NoNNR) or the ReLU cosine gate (NoRCG) leads to severe degradation in reconstruction fidelity or inflated composition scores. Table 15 reports the sensitivity of $C^2R$ to the regularization strength $\lambda_{C^2R}$, showing that $\lambda_{C^2R}$=5 achieves the best trade-off.

## I. Compatibility with Other SAE Architectures

Table 16 reports the full results of applying $C^2R$ to three different SAE backbones: TopK, OrtSAE, and AbsTopK. In each pair, adding $C^2R$ improves all structural metrics without degrading reconstruction fidelity, confirming that $C^2R$ is a backbone-agnostic regularizer.

## J. Downstream Causal Intervention Tasks

Table 17 reports the results on two causal intervention tasks from SAEBench. $C^2R$ achieves the second-best performance on both SCR and TPP, outperforming TopK, Batch TopK, and Ort, and achieving comparable results to Matryoshka.

## K. Empirical Verification of Eq. 13

We verify the condition in Eq. 13 using our trained SAEs on Gemma-2-2B layer 12 with a 4M-token test set from SAEBench.

Figure 4 shows the distribution of the log A/B ratio, i.e., $\log \frac{\sum_{i=1}^{m}(z_1^{(i)})^2}{\sum_{i=m+1}^{m+n}(z_2^{(i)})^2}$, across all absorption pairs. The median A/B ratio is 81.56, confirming that the cumulative energy of the primary feature strongly dominates the residual in practice.

Figure 5 shows the scatter plot of $\alpha$ versus the right-hand side of Eq. 13 for all absorption pairs. Points above the diagonal satisfy the condition. 88.1% of pairs ($N$=4,555) fall in the satisfied region, and the violated pairs are concentrated in a low-$\alpha$ regime where absorption is minimal.

## L. Relationship to OrtSAE

The chunk-wise nearest-neighbor calculation in Appendix B is an engineering optimization to reduce complexity and control variables, not a methodological dependence. Fundamentally, OrtSAE constrains decoder weights, whereas $C^2R$ operates in activation space using the Minkowski inequality. In $C^2R$, decoder cosine similarity acts as a gating weight ($\rho^2$) to target redundant features, creating a gradient dynamically scaled by activation magnitude $S$. This adaptive approach distinguishes $C^2R$ from OrtSAE's uniform penalty, protecting low-frequency features from premature destruction. As shown in Table 16,

| Method | KL Score ↑ | Interp ↑ | Composition ↓ | Absorption ↓ | Split ↓ |
|--------|-----------|----------|---------------|--------------|---------|
| Batch TopK | 0.9606 | 0.9211 | 0.3227 | 0.1920 | 1.1538 |
| Matryoshka | 0.9602 | 0.9117 | 0.1781 | **0.0158** | **1.0385** |
| Ort | 0.9611 | 0.9202 | 0.1053 | 0.0627 | 1.0769 |
| C$^2$R | **0.9618** | **0.9424** | **0.0766** | 0.0396 | **1.0385** |

Table 13. Results on The Pile (Gemma-2-2B, layer 12, $k$=100).

| Method | KL Score ↑ | Interp ↑ | Composition ↓ | Absorption ↓ | Split ↓ |
|--------|-----------|----------|---------------|--------------|---------|
| Batch TopK | 0.9598 | 0.9208 | 0.3321 | 0.1985 | 1.1154 |
| Ort | 0.9617 | 0.9208 | 0.1046 | 0.0606 | 1.1154 |
| C$^2$R (NoNNR) | 0.9439 | 0.9488 | 0.5655 | **0.0255** | 1.1538 |
| C$^2$R (NoRCG) | 0.8439 | **0.9873** | 0.4326 | 0.0520 | 1.1923 |
| C$^2$R | **0.9629** | 0.9239 | **0.0990** | 0.0590 | **1.0769** |

Table 14. Component ablation. NoNNR: without the nearest-neighbor restriction. NoRCG: without the ReLU cosine gate. Removing either component causes a severe degradation in reconstruction fidelity or feature composition.

C$^2$R can be applied on top of OrtSAE to achieve further improvements, confirming that the two methods are complementary.

## M. Sensitivity to Feature Frequency

Since C$^2$R aggregates batchwise $\ell_2$ norms, the effective regularization strength on a given latent pair scales with how frequently those features appear in the batch. To investigate whether rare features receive insufficient regularization, we analyze the relationship between feature frequency and absorption rate using 26 letter-specific features from SAEBench on a Gemma-2-2B Batch TopK SAE.

As shown in Figure 6, the Pearson correlation between feature frequency and absorption rate is $r = +0.301$ ($p = 0.135$), which is not statistically significant. The Spearman correlation is $\rho = +0.403$ ($p = 0.041$), suggesting a slight positive trend where absorption may marginally increase with frequency rather than decrease. Rare features such as Q, J, and X exhibit absorption rates below 0.1, comparable to or lower than high-frequency features such as C and S. This indicates that the consistency pressure remains effective across the frequency spectrum and that rare features do not suffer from reduced reliability due to batch statistics.

## N. Dead Feature Rates

Table 18 reports the dead feature rates for all methods on Gemma-2-2B layer 12. C$^2$R maintains high dictionary utilization with a dead feature rate under 1%.

| $\lambda_{C^2R}$ | KL Score ↑ | Interp ↑ | Composition ↓ | Absorption ↓ | Split ↓ |
|---|---|---|---|---|---|
| – (Batch TopK) | 0.9598 | 0.9196 | 0.3321 | 0.1985 | 1.1154 |
| – (Ort) | 0.9617 | 0.9196 | 0.1046 | 0.0606 | 1.1154 |
| 0.1 | 0.9629 | 0.9291 | 0.2944 | 0.1926 | **1.0385** |
| 0.5 | **0.9638** | **0.9496** | 0.2138 | 0.1726 | **1.0385** |
| 1 | 0.9633 | 0.9407 | 0.1676 | 0.1349 | 1.1154 |
| 5 | 0.9629 | 0.9239 | 0.0990 | 0.0590 | 1.0769 |
| 10 | 0.9521 | 0.9168 | **0.0777** | **0.0425** | **1.0385** |

*Table 15.* Sensitivity to $\lambda_{C^2R}$. Baselines (Batch TopK and Ort) are shown above the rule for reference. $\lambda_{C^2R}{=}5$ achieves the best trade-off between reconstruction fidelity and structural metrics.

| Method | KL Score ↑ | Interp ↑ | Composition ↓ | Absorption ↓ | Split ↓ |
|---|---|---|---|---|---|
| TopK | 0.9620 | 0.8762 | 0.3311 | 0.2680 | 1.3462 |
| TopK + C²R | **0.9627** | **0.9227** | **0.1004** | **0.0752** | **1.0769** |
| Ort | 0.9617 | 0.9196 | 0.1046 | 0.0606 | 1.1154 |
| Ort + C²R | **0.9623** | **0.9217** | **0.0662** | **0.0524** | **1.0385** |
| AbsTopK | 0.9576 | 0.6759 | 0.2357 | 0.1407 | 1.3462 |
| AbsTopK + C²R | **0.9580** | **0.7579** | **0.0810** | 0.0846 | **1.0769** |

*Table 16.* C²R applied to different SAE backbones (Gemma-2-2B, layer 12, $k{=}100$). Within each pair, the better value is in **bold**. C²R consistently improves structural metrics across all architectures.

| Method | SCR ↑ | TPP ↑ |
|---|---|---|
| TopK | 0.0481 | 0.0038 |
| Batch TopK | 0.0353 | 0.0045 |
| Matryoshka | **0.1172** | **0.0370** |
| Ort | 0.0859 | 0.0061 |
| C²R | 0.1047 | 0.0226 |

*Table 17.* Downstream causal intervention tasks (Gemma-2-2B, layer 12, $k{=}100$). SCR: Spurious Correlation Removal. TPP: Targeted Probe Perturbation. **Bold** and underline indicate the best and second-best values, respectively.

| Method | Dead Feature Rate ↓ |
|---|---|
| TopK | 0.11% |
| Batch TopK | **0.06%** |
| Matryoshka | 0.10% |
| Ort | 4.61% |
| C²R | 0.90% |

*Table 18.* Dead feature rates on Gemma-2-2B layer 12. C²R maintains over 99% dictionary utilization.

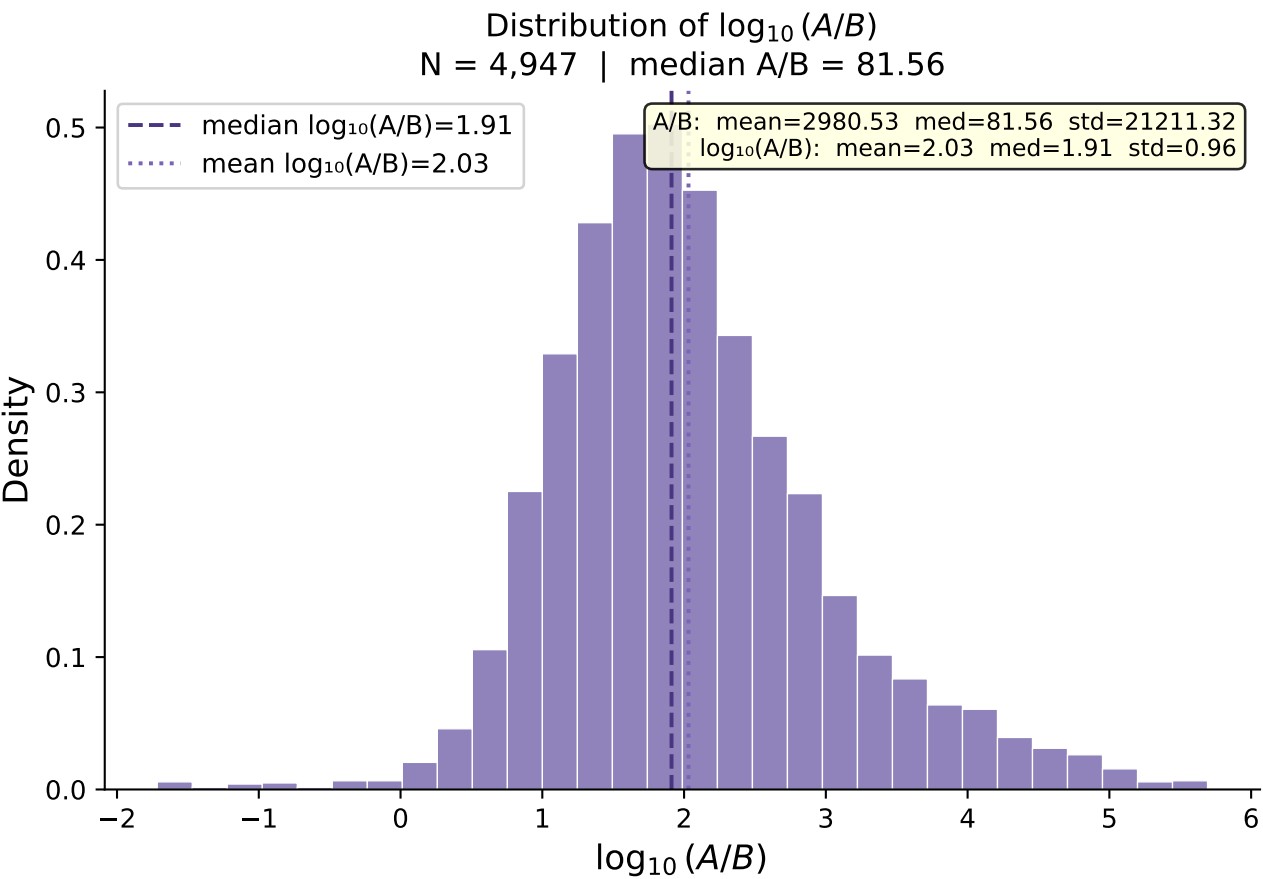

*Figure 4.* Distribution of the log A/B ratio across absorption pairs ($N$=4,947, median = 81.56).

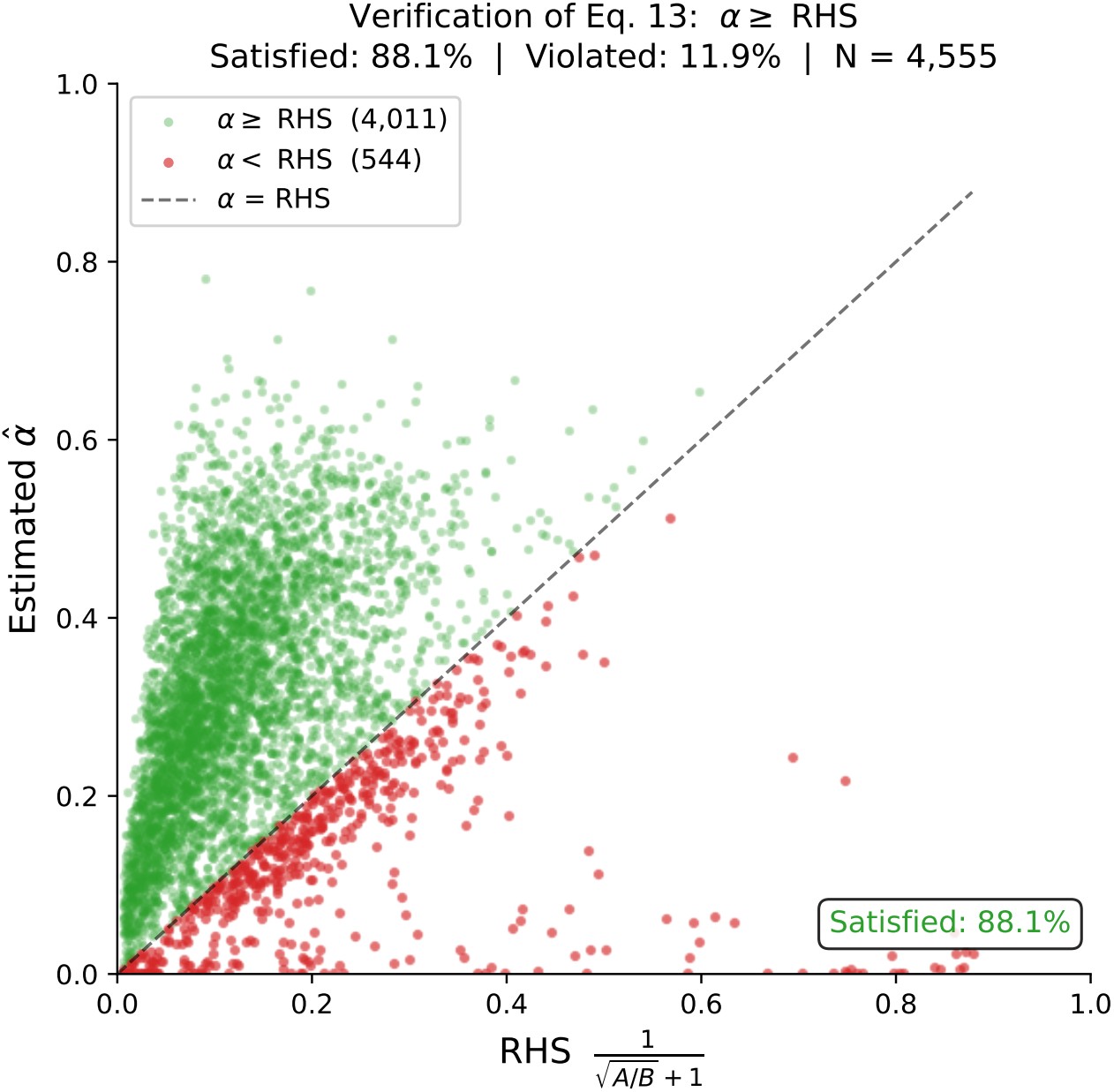

*Figure 5.* Scatter plot of $\alpha$ vs. RHS of Eq. 13 ($N$=4,555). 88.1% of pairs satisfy the condition.

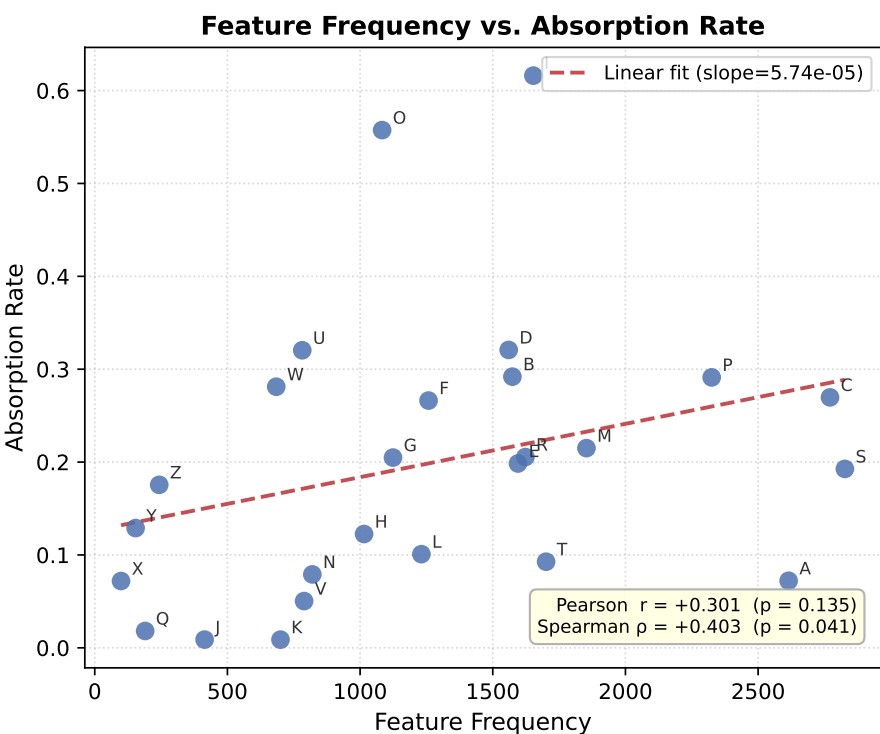

*Figure 6.* Absorption rate vs. feature frequency for 26 letter-specific features (Gemma-2-2B, Batch TopK SAE). No significant negative correlation is observed, indicating that rare features do not suffer from higher absorption.

