# OpenReview forum: "C$^{2}$R: Cross-sample Consistency Regularization Mitigates Feature Splitting and Absorption in Sparse Autoencoders"
_ICML.cc/2026/Conference — ICML 2026 regular_

### Official Review · Reviewer_B3rG · 2026-03-04

**Soundness:** 2
**Presentation:** 2
**Significance:** 2
**Originality:** 3
**Overall Recommendation:** 4
**Confidence:** 3

**Summary:**

Sparse Autoencoders (SAEs) are gaining significant attention as a useful tool that maps complex latent factors within LLMs into interpretable features by imposing sparsity in the encoded variables. However, conventional approaches that employ $\ell_1$ or $TopK$ penalty as the regularization method are shown to induce feature splitting or feature absorption, particularly when dealing with latent factors that possess an inherent hierarchy. To mitigate these artifacts, this paper proposes incorporating the Cross-sample Consistency Regularization (C2R) penalty, which takes a form of $\ell_{2,1}$ group norm penalty adjusted with the redundancy parameter $\alpha$. $\alpha$ denotes ''leakage'' of the latent feature vector onto its orthogonal basis. Adding C2R penalty to the conventional SAE loss prevents total feature splitting or absorption of the ideal latent vector the into its residual in learning the dictionary of SAEs. To validate the effectiveness of the proposed penalty, the authors compared various metrics between SAE models trained with and without C2R penalty on an open text dataset.

**Compliance With Llm Reviewing Policy:**

Affirmed.

**Final Justification:**

Authors have adequately addressed my concerns, and I have raised my score.

**Key Questions For Authors:**

1. Regarding to the Weaknesses section, how would C2R penalty perform if there is a significant imbalance between the ideal case (ratio of $m$ and $n$) and the actual test data about the hierarchy of the features?

1. In practice, how should the sparsity level $k$ and the hyperparameter $\lambda_{C^2R}$ should be selected?

1. How sensitive is the proposed C2R regularization to the choice of the underlying SAE model? Does its effect particularly differs by the base SAE models?

**Limitations:**

Yes

**Strengths And Weaknesses:**

Strengths

* Authors have well addressed the intuition behind the C2R penalty by illustrating the ideal representation and demonstrating how conventional approaches collapse this structure.

* Empirical results demonstrate that the proposed penalty successfully mitigates feature splitting and absorption without significantly compromising reconstruction quality.

Weaknesses

*  Authors claim they have integrated C2R into four different SAE architectures, but there is only one plot that is marked as C2R in Figure 2 and Figure 3, hence it makes unclear which specific model architectures were used for presenting this result.

*  In constructing C2R, authors assume an ideal scenario where leakage occurs in $n$ out of $m + n$ batch samples, and this ratio appears to remain fixed during the actual training. While C2R method shows better Feature absorption and Feature splitting metrics, these metrics proposed requires test data equipped with ground-truth labels about the feature hierarchy in a balanced ratio. I am concerned that C2R penalty might inadvertently reflect or exploit this predetermined ratio information when choosing the size of $m$ and $n$. Refer to the question section.

---

> ### Author Rebuttal · Authors · 2026-03-31
>
> Thank you for your thorough review and constructive feedback. Below are our responses to your comments regarding architecture integration, model sensitivity, theoretical assumptions, and hyperparameter selection. In the following tables, the best and second-best values for each metric are marked in **bold** and *italics*, respectively.
>
> **W1: Integration into Multiple SAE Architectures**
>
> > "Authors claim they have integrated $C^{2}R$ into four different SAE architectures, but there is only one plot that is marked as $C^{2}R$ in Figure 2 and Figure 3, hence it makes unclear which specific architecture was used."
> >
>
> We sincerely apologize for this error. In our initial experiments, we did integrate $C^{2}R$ into some different SAE architectures; however, due to space constraints, we were unable to present all of the results. For the main results presented in Figures 2 and 3, we chose to use **Batch TopK** as the base architecture to ensure a fair and direct comparison with OrtSAE and Matryoshka SAE, both of which also utilize Batch TopK as their underlying backbone.
>
> **W2: Sensitivity to the Underlying SAE Model and Scaling**
>
> > "How sensitive is the proposed $C^{2}R$ regularization to the choice of the underlying SAE model? Does its effect particularly differs by the base SAE models?"
> >
>
> To address the concern regarding model sensitivity, we applied $C^{2}R$ to both the standard **TopK** and **OrtSAE** backbones. The results in the table below demonstrate that $C^{2}R$ provides significant improvements in structural metrics (Composition, Absorption, and Splitting) regardless of the base model, confirming that its benefits are not tied to a specific architecture.
>
> | Method | **KL Score↑** | **Interp↑** | **Composition↓** | **Absorb↓** | **Split↓** |
> | --- | --- | --- | --- | --- | --- |
> | TopK | 0.962 | 0.8762 | 0.3311 | 0.268 | 1.3462 |
> | **TopK + $C^{2}R$** | **0.9627** | **0.9227** | *0.1004* | 0.0752 | *1.0769* |
> | Ort | 0.9617 | 0.9196 | 0.1046 | *0.0606* | 1.1154 |
> | **Ort + $C^{2}R$** | *0.9623* | *0.9217* | **0.0662** | **0.0524** | **1.0385** |
>
> We also evaluated the generalizability of $C^{2}R$ on larger models, Qwen3-8B and Llama-3-8B, at layer 20. $C^{2}R$ consistently achieves optimal or near-optimal results on structural feature metrics while maintaining reconstruction fidelity. **For the result tables, please refer to the W1 section of our response to Reviewer k9Se.**
>
>
>
> **W3: Empirical Assumptions about $m$ and $n$**
>
> > "I am concerned that $C^{2}R$ penalty might inadvertently suppress legitimate features... Regarding to the Weaknesses section, how would $C^{2}R$ penalty perform if there is a significant imbalance between the ideal case... and the actual test data...?"
> >
>
> We appreciate your concern regarding the ratio of $m$ (samples where only the "pure" feature appears) and $n$ (samples where features are split or absorbed). To verify this assumption, we collected statistics on feature pairs exhibiting absorption over 2M tokens. Our results (viewable at https://imgur.com/a/NXe1rT6) show that $m$ is significantly larger than $n$ in practice.
>
> Furthermore, we empirically tested the conditions for the theoretical guarantees in Eq. 13 and Eq. 14. Let the Left-Hand Side (LHS) and Right-Hand Side (RHS) of Eq. 14 be A and B, respectively; our measurements on Batch TopK SAEs show the median ratio of A/B reaches 75 (distribution: https://imgur.com/a/hRKSUnh). Additionally, we found that the condition in Eq. 13 is satisfied in 81.8% of cases (https://imgur.com/a/IMpt3nF). These findings provide a solid empirical basis for the theoretical consistency pressure derived in the paper.
>
> **W4: Hyperparameter Selection ($k$ and $\lambda_{C^{2}R}$)**
>
> > ".. how should the sparsity level k and the hyperparameter $\lambda_{C^{2}R}$ should be selected?"
> >
>
> For a given SAE model, the sparsity level $k$ is typically chosen based on the desired "Loss Recovered" metric, usually targeting a range near 1.0. Regarding $\lambda_{C^{2}R}$, we recommend selecting the largest value that does not significantly degrade reconstruction fidelity. In our experiments, we found that $\lambda_{C^{2}R}=5$ provides the best balance, as shown in the sensitivity table below:
>
> | Method | **KL Score↑** | **Interp↑** | **Composition↓** | **Absorb↓** | **Split↓** |
> | --- | --- | --- | --- | --- | --- |
> | Batch TopK | 0.9598 | 0.9196 | 0.3321 | 0.1985 | 1.1154 |
> | Ort | 0.9617 | 0.9196 | 0.1046 | 0.0606 | 1.1154 |
> | lambda_$C^{2}R$=0.1 | 0.9629 | 0.9291 | 0.2944 | 0.1926 | **1.0385** |
> | lambda_$C^{2}R$=0.5 | **0.9638** | **0.9496** | 0.2138 | 0.1726 | **1.0385** |
> | lambda_$C^{2}R$=1 | *0.9633* | *0.9407* | 0.1676 | 0.1349 | 1.1154 |
> | lambda_$C^{2}R$=5 | 0.9629 | 0.9239 | *0.099* | *0.059* | *1.0769* |
> | lambda_$C^{2}R$=10 | 0.9521 | 0.9168 | **0.0777** | **0.0425** | **1.0385** |
>
> We will incorporate the above discussions and results into the final version of the manuscript.

---

> > ### Author Rebuttal · Reviewer_B3rG · 2026-04-02
> >
> > I thank the authors for their thorough responses to my questions. I am still concerned whether the proposed training method promoting the information leakage is appropriate. From what I have read, I am assuming that fixed ratio of $m:n$ is used to compute the regularization term $\mathcal{L}_{\text{pair}}$ in C2R loss for each batch (correct me if I am wrong). However, your results in (https://imgur.com/a/NXe1rT6) show that the proportion of $n$ barely reaches 10\%. How was the ratio of $m:n$ in the training stage was chosen? Wouldn't the significant mismatch of the appropriate $m:n$ between in the training and the test data be a problem?

---

> > > ### Author Response · Authors · 2026-04-03
> > >
> > > **Dear Reviewer B3rG,**
> > >
> > > Thank you for the follow-up. We would like to clarify these details directly:
> > >
> > > **1. Regarding the $m:n$ ratio and information leakage:**
> > >
> > > **We do not use a "fixed" or "chosen" $m:n$ ratio.** $m$ and $n$ are not hyperparameters, nor are they used to compute the actual $C^2R$ loss in practice. Therefore, our method does not promote information leakage. These variables are strictly analytical notations introduced in Sections 3 and 4 to theoretically prove why the loss function is effective.
> > >
> > > **2. Regarding the potential train/test mismatch:**
> > >
> > > **In our setup, a "significant mismatch" does not occur.** Within the same dataset, the overlap fraction $n/(m+n)$ demonstrates strong stability. Specifically, our empirical results (https://imgur.com/a/NXe1rT6) show a consistently low mean of 1.6% with a minimal standard deviation of 0.052. Furthermore, we tested if the $m:n$ ratio changes across different datasets (OpenWebText, The Pile, and FineWeb). The results (see the distribution plots at [https://imgur.com/a/2iAvr9z](https://www.google.com/search?q=https://imgur.com/a/2iAvr9z)) show it is very stable: the mean overlap fraction $n/(m+n)$ consistently stays around 1.1% to 1.6%. A significant mismatch between train and test data is therefore unlikely in practice even if in out-of-distribution scenarios.
> > >
> > > We will ensure these points are stated clearly in the final manuscript. We hope this addresses your concerns.

---

### Official Review · Reviewer_9euN · 2026-03-10

**Soundness:** 3
**Presentation:** 3
**Significance:** 3
**Originality:** 3
**Overall Recommendation:** 5
**Confidence:** 4

**Summary:**

Standard SAE sparsity objectives (ℓ₁ and TopK) operate on individual samples and consequently provide no mechanism to ensure that a single semantic feature is consistently represented by the same latent across inputs. This paper formalizes feature splitting and feature absorption as a unified failure mode parameterized by a scalar α that quantifies cross-latent feature leakage, then proves via the Minkowski inequality that per-sample objectives actively incentivize high α. The proposed C²R regularization penalizes the batchwise sum of ℓ₂ norms for the nearest-neighbor pair of each latent, gated by squared rectified cosine similarity between decoder vectors, thereby selectively targeting redundant latent pairs without collapsing independent features. Experiments on Gemma-2-2B residual stream activations using 65,536-latent dictionaries show reduced splitting and absorption relative to all reconstruction-preserving baselines, with no degradation in Loss Recovered, KL Divergence, AutoInterp, or disentanglement scores.

**Compliance With Llm Reviewing Policy:**

Affirmed.

**Final Justification:**

The authors have addressed all my concerns with sufficient empirical evidence. They were very responsive throughout the rebuttal period. As all my main concerns have been addressed, I am recommending Accept.

**Key Questions For Authors:**

**1. Verification of the theoretical guarantee's condition.** The monotonicity result in Appendix A holds only when the condition in Eq. 13 is satisfied, which requires $\sum_{i=1}^{m}(z_1^{(i)})^2 \gg \sum_{i=m+1}^{m+n}(z_2^{(i)})^2$. This is justified by citing frequency and magnitude statistics from external papers rather than from your own trained SAEs. Can you provide an empirical distribution of the right-hand side of Eq. 13 across latents in your trained 65,536-latent dictionaries, and characterize what fraction of latents violate the condition?

**2. Isolation of C²R's contribution from the Batch TopK backbone.** All reported results compare C²R-enhanced Batch TopK against vanilla baselines, making it impossible to determine how much of the gain is attributable to C²R alone versus the relaxed per-sample sparsity constraint of Batch TopK. Can you provide results for C²R applied on top of at least one additional backbone, such as vanilla TopK?

**3. Sensitivity to batch composition and the treatment of rare features.** Since $L_{C^2R}$ aggregates batchwise ℓ₂ norms $\|Z_{:,i}\|_2$, the effective regularization strength on any given latent pair scales with how frequently those features appear in the batch. Rare features, which are precisely those most susceptible to absorption according to Chanin et al. (2024), will contribute small $S_{i,j^*(i)}$ values and may receive negligible regularization pressure. Did you observe differences in absorption rates between high-frequency and low-frequency features?

**Limitations:**

The limitations section acknowledges the restriction to Gemma-2-2B and the absence of downstream task evaluation, which is appropriate. Three gaps should be added. First, the conditional nature of the theoretical guarantee (Eq. 13) should be stated explicitly in the main text rather than implied by a derivation in Appendix A; Table 1's unconditional checkmark overstates what is proven. Second, the use of a non-public LLM judge for the AutoInterp evaluation should be flagged as a reproducibility limitation. Third, the paper should discuss the conditions under which C²R may suppress legitimate polysemanticity, specifically when genuinely distinct features have moderately aligned decoder directions, and the potential for the regularization to silently reduce dictionary utilization by increasing dead feature rates.

**Strengths And Weaknesses:**

### Soundness
The two-latent theoretical analysis is carefully constructed. Lemma 4.1 is valid: the Minkowski inequality strictly establishes that ℓ₁(α > 0) < ℓ₁(α = 0) for any α ∈ (0, 1], and the TopK sparsity budget argument (Eqs. 10 and 11) is correct. The gradient decomposition in Eq. 18 into a consistency pressure and an adaptive orthogonality pressure is insightful and correctly derived. However, the theoretical guarantee depends on the condition in Eq. 13, which requires the cumulative energy of the primary feature to dominate the residual across the batch (Eq. 14). This condition is justified by appeal to findings from Leask et al. and Chanin et al. rather than from the authors' own training runs, making the guarantee conditional rather than universal. Table 1 presents C²R as offering an unconditional theoretical guarantee against both splitting and absorption, which overstates what is proven. Additionally, the ReLU gate in Eq. 16 renders C²R blind to near-antipodal redundancy where two decoder directions satisfy $\langle \hat{w}_i, \hat{w}_j \rangle < 0$, a scenario that can arise in signed or symmetric encoder settings. No statistical validation across random seeds is reported, and all results reflect single training runs.

### Presentation
The paper is well-organized. The progression from unified formalism (Section 3) through theoretical pathology analysis (Section 4) to method derivation (Section 5) and experiments (Section 6) is logical and readable. Figure 1 communicates the intuition effectively. Several reproducibility concerns are significant, however. The AutoInterp evaluation depends on GPT-5-mini, a model not publicly available at submission time, making this metric entirely non-reproducible externally. The hyperparameter selection of $\lambda_{C^2R} = 5$ is described only as "maximal regularization strength that does not degrade reconstruction fidelity," with no performance curve across the swept values {0.1, 0.5, 1, 5, 10} provided. The condition in Eq. 13, which is load-bearing for the main theoretical claim, is derived in Appendix A but receives insufficient treatment in the main body, where it is mentioned only briefly before the claim is asserted.

### Significance
C²R addresses a well-documented and practically consequential problem. Feature absorption and splitting directly undermine causal analysis, circuit discovery, and model steering, which are core applications motivating SAE research. The method is computationally practical, orthogonal to encoder-architecture improvements such as Gated and JumpReLU SAEs, and potentially stackable with them. The finding that Matryoshka SAEs achieve low splitting and absorption only at the cost of reconstruction fidelity is a useful clarifying result for the field. The significance is meaningfully constrained, however, by the narrow experimental scope: a single model, a single layer, a single dataset, and no downstream task evaluation. Leask et al. (cited) establish that splitting scales with model size, making the absence of any larger-model experiment a genuine gap for a method motivated in part by scaling concerns.

### Originality
The unification of feature splitting and absorption under a single α-parameterized geometric framework is an original conceptual contribution; prior work treats these as empirically distinct phenomena. The selective gating of the Minkowski penalty by decoder cosine similarity to distinguish redundant from independent latent pairs is a non-obvious and principled design choice. The adaptive scaling of orthogonality pressure by activation magnitude S (the $S\nabla\rho^2$ term in Eq. 18), which protects rare and emerging latents from premature orthogonality enforcement, represents a meaningful improvement over the uniform penalty used in OrtSAE. The relationship to OrtSAE nevertheless warrants more careful treatment: the computational implementation in Appendix B is explicitly adopted from OrtSAE, and the orthogonality pressure component of C²R's gradient is closely related in goal. The paper does not compare against AbsTopK (signed TopK), which targets similar symptoms through a different architectural mechanism and is directly relevant to understanding whether the ReLU cosine gate remains appropriate in signed encoder settings.

---

> ### Author Rebuttal · Authors · 2026-03-31
>
> Thank you for your thorough review and constructive feedback.
>
> **W1 & Q1: Verification of the theoretical guarantee's condition (Eq. 13 and 14)**
>
> We empirically verified the conditions in Eq. 13 and Eq. 14 using our trained SAEs. On a 2M-token test set via SAEBench, the median A/B ratio (LHS vs. RHS of Eq. 14) is 75, confirming the primary feature's dominance (https://imgur.com/a/hRKSUnh). The exact condition in Eq. 13 is satisfied in 81.8% of cases (https://imgur.com/a/IMpt3nF). We will explicitly state the conditional nature of this guarantee in the main text and include these empirical distributions in the revised appendix.
>
> **W2: The ReLU gate mechanism in Eq. 16 & Comparison against Abs TopK**
>
> To address concerns regarding signed encoders, we evaluated $C^{2}R$ on top of Abs TopK (Gemma-2-2B layer 12). As shown in this table (see https://imgur.com/a/MwyNyYT), $C^{2}R$ effectively adapts to bi-directional architectures, significantly improving feature hierarchy while preserving reconstruction fidelity.
>
> The ReLU gate (Eq. 16)  is a deliberate choice. Per the definitions in Section 3, feature absorption and splitting occur when child features capture projections of orthogonal parent features, which mathematically implies a cosine similarity $> 0$. By targeting only positive similarities, we avoid "collateral damage" to non-pathological negative correlations in signed settings, ensuring training stability.
>
> **W3: Statistical validity across random seeds**
>
> Please refer to our response to Reviewer WPYq (specifically the W1 & Q1 section), where we have provided a comprehensive table with 95% confidence intervals and multi-run experimental results.
>
> **W4: Reliance on GPT-5-mini for evaluation**
>
> While newer versions are the default, the gpt-5-mini model can generally still be accessed via the OpenAI API or platforms like OpenRouter. We also re-evaluated the AutoInterp scores using gpt-5. The scores and high Pearson correlations (p=0.0) are presented in this table (see https://imgur.com/a/IBxqRNb).
>
> Appendix D and Table 6 provide the correlation between GPT-5-mini scores and human evaluations, which is also extremely high. We will incorporate this discussion into Section 6.5 and the Appendix in the final version of the manuscript.
>
> **W5: Selection of $\lambda_{C^{2}R}=5$**
>
> Please refer to the W3 section of our response to Reviewer WPYq.
>
> **W6: Narrow experimental scope (Single model, layer, and dataset)**
>
> Please refer to the W1 and W2 sections of our response to Reviewer k9Se.
>
> **W7: More discuss about the relationship to OrtSAE**
>
> The chunk-wise calculation in Appendix B is an engineering optimization to reduce complexity and control variables, not a methodological dependence. Fundamentally, OrtSAE constrains decoder weights, whereas $C^{2}R$ operates in activation space using the Minkowski inequality. In $C^{2}R$, decoder cosine similarity acts as a gating weight ($\rho^2$) to target redundant features, creating a gradient dynamically scaled by activation magnitude $S$. This adaptive approach distinguishes $C^{2}R$ from OrtSAE's uniform penalty, protecting low-frequency features from premature destruction. We will add this discussion to Section 5.2 and the Appendix.
>
> **Q2: Isolation of $C^{2}R$'s contribution from the Batch TopK backbone**
>
> Please refer to the W2 section of our response to Reviewer B3rG.
>
> **Q3: Sensitivity to batch composition and rare features**
>
> Per your suggestion, we analyzed the relationship between feature frequency and absorption rate using 26 letter-specific features from SAEBench on a Gemma-2-2b Batch TopK SAE. We found no statistically significant evidence that low-frequency features suffer from higher absorption rates (https://imgur.com/a/sGGeZjc). The Pearson correlation is $r = +0.301$ ($p = 0.135$), exceeding the 0.05 threshold. The Spearman correlation ($\rho = +0.403, p = 0.041$) suggests a slight positive trend, implying absorption may marginally increase with frequency. Rare features (e.g., Q, J, X) exhibit absorption rates below 0.1, comparable to or lower than high-frequency features like C and S. This indicates consistency pressure remains effective across the frequency spectrum and rare features do not lack reliability due to batch statistics.
>
> **Limitations: Three gaps should be added**
>
> We appreciate these constructive suggestions. In the final manuscript, we will explicitly state the conditional nature of the theoretical guarantee in the main text and update Table 1 with conditional markers. Additionally, we will incorporate the discussion about GPT-5-mini as a judge (as discussed in W4 above) into Section 6.5. Finally, we will include a discussion on the potential for $C^{2}R$ to suppress legitimate polysemanticity or increase dead feature rates when genuinely distinct features have moderately aligned decoder directions.
>
> We will incorporate the above discussions and results into the final version of the manuscript.

---

> > ### Author Rebuttal · Reviewer_9euN · 2026-04-03
> >
> > Thank you for the detailed rebuttal. The five-run statistical validation with 95% confidence intervals, the cross-backbone ablations, the extension to Qwen3-8B and Llama-3-8B, the downstream task results on SCR and TPP, the empirical verification of Eq. 13 and 14, the AbsTopK experiment, and the $\lambda_{C^2R}$ sensitivity table collectively address the most substantial concerns raised in my review.
> >
> > Two points remain partially open. The 18.2% violation rate of the condition in Eq. 13 is noted but not characterized. It would strengthen the paper to briefly identify whether these latents correspond systematically to residual absorption cases. Additionally, no dead feature statistics are reported despite the commitment to discuss this. I ask that these be included in the final version.
> >
> > Notwithstanding these minor gaps, the rebuttal demonstrates that C²R's gains are statistically robust, backbone-agnostic, and generalizable across models, layers, and datasets, and that the method delivers measurable improvements on downstream causal intervention tasks. I am raising my recommendation to Accept.

---

> > > ### Author Response · Authors · 2026-04-05
> > >
> > > **Dear Reviewer 9euN,**
> > >
> > > Thank you for the follow-up. We would like to clarify these details directly:
> > >
> > > **1. Regarding the 18.2% violation rate (Eq. 13):**
> > >
> > > **The 18.2% violated pairs systematically correspond to marginal cases characterized by very weak absorption and low primary feature energy.** We conducted a systematic analysis of all absorption pairs, and the results are as follows:
> > >
> > > | **Metric** | **Satisfied (81.8%)** | **Violated (18.2%)** |
> > > | --- | --- | --- |
> > > | **Mean α** | 0.317 | 0.127 |
> > > | **Median α** | 0.311 | 0.124 |
> > > | **Mean RHS** | 0.117 | 0.228 |
> > > | **Median RHS** | 0.103 | 0.195 |
> > >
> > > The violated group exhibits two systematic patterns:
> > >
> > > **Minimal Absorption:** The mean α for violated pairs is only 0.127, indicating that the child feature absorbs a negligible fraction of the parent's signal.
> > >
> > > **Weak Primary Features:** The RHS mean (0.228) is much higher than in the satisfied group (0.117). This indicates that the parent feature's accumulated energy in isolated activations is relatively small compared to the child feature's energy in overlapping activations.
> > >
> > > This demonstrates that these violations do not correspond to the severe absorption cases. Instead, they are marginal instances, confirming that the theoretical guarantee of $C^2R$ covers the vast majority of significant absorption behaviors.
> > >
> > > **2. Regarding the dead feature rates:**
> > >
> > > **We evaluated the dead feature rates and found that $C^2R$ maintains high dictionary utilization.** Specifically, we measured the dead feature rates for SAEs on Gemma-2-2B layer 12:
> > >
> > > | **Method** | **Dead Feature Rates ↓** |
> > > | --- | --- |
> > > | TopK | 0.11% |
> > > | Batch TopK | **0.06%** |
> > > | Matryoshka | *0.10%* |
> > > | Ort | 4.61% |
> > > | $C^2R$ | 0.90% |
> > >
> > > As shown, $C^2R$ maintains a dead feature rate of under 1% (0.90%). While slightly higher than the Batch TopK baseline, it utilizes over 99% of the dictionary latents, confirming that the regularization does not cause significant dictionary capacity waste.
> > >
> > > We will ensure these empirical details are stated clearly in the final manuscript.
> > >
> > > Finally, we truly appreciate your note about raising the recommendation to "Accept." We noticed that the system currently still displays "Weak Accept," so we just wanted to leave a gentle reminder here in case the dropdown menu needs to be manually updated on your end. Thank you again for your time and constructive feedback to help us strengthen this paper.

---

### Official Review · Reviewer_k9Se · 2026-03-13

**Soundness:** 3
**Presentation:** 3
**Significance:** 4
**Originality:** 4
**Overall Recommendation:** 5
**Confidence:** 4

**Summary:**

This paper studies two failure modes in sparse autoencoders for LLM interpretability, feature splitting and feature absorption, and argues that both arise because standard per-sample sparsity objectives do not enforce consistent latent representations across examples. To address this, the authors propose C2R, a batch-level regularization method that encourages semantically similar information to be captured by a single latent rather than split across redundant features. Extensive experiments on Gemma-2-2B activations show that C2R reduces splitting, absorption, and redundancy compared with several SAE baselines while largely preserving reconstruction quality and interpretability.

**Compliance With Llm Reviewing Policy:**

Affirmed.

**Final Justification:**

Thank you to the authors for providing additional details on the follow-up experiments. After carefully reviewing the results, I would like to keep the current score unchanged.

**Key Questions For Authors:**

- How does C2R perform when scaling to larger model sizes and substantially larger SAE training corpora? Since the current experiments are limited to Gemma-2-2B and a 500M-token OpenWebText subset, evidence that the method continues to improve splitting and absorption at larger scale would strengthen my confidence in its broader applicability; if the gains diminish or become unstable, that would suggest the method may be more setting-specific.

**Limitations:**

yes

**Strengths And Weaknesses:**

## Strengths
- The C2R method is novel: it introduces a batch-level cross-sample consistency regularizer that penalizes co-activation of directionally similar latents, with a theoretical motivation based on the Minkowski inequality and a selective implementation using decoder cosine similarity to identify likely redundant latent pairs.
- Extensive benchmarks are included across several SAE baselines, including TopK SAE, Batch TopK SAE, Matryoshka SAE, and OrtSAE, and across multiple sparsity levels k ∈ {60, 80, 100, 120}.
- The evaluations are fairly comprehensive: the paper measures reconstruction fidelity with Loss Recovered and KL Div. Score, interpretability with AutoInterp, feature structure with Split Num, Absorption Rate, and Composition, and disentanglement with the RAVEL benchmark.
- The empirical results cover both the paper’s main claimed benefits and standard SAE quality metrics, showing improvements on feature splitting, feature absorption, and feature composition, while largely maintaining reconstruction, autointerp, and disentanglement relative to strong baselines.

## Weakness
- All experiments are conducted only on Gemma-2-2B, primarily on the layer-12 residual stream, which is still a relatively small setting by current LLM standards. It would strengthen the paper to test whether C2R continues to help on larger models or in a broader range of architectures and layers.
- The SAE training setup uses only a 500M-token subset of OpenWebText, which feels somewhat limited for establishing robustness. Since OpenWebText is a broad web-text corpus collected from URLs shared on Reddit, rather than a controlled set of categories, it would be helpful to better understand how sensitive the results are to the choice, scale, and composition of the training data.

---

> ### Author Rebuttal · Authors · 2026-03-31
>
> Thank you for your thorough review and constructive feedback. In the tables below, the best and second-best values for each metric are marked in **bold** and *italics*, respectively.
>
> Below are our detailed responses to your questions and concerns:
>
> **W1: Broader Range of Architectures and Layers**
>
> > *"How does $C^{2}R$ perform when scaling to larger model sizes...?"* / *"...evaluate on a broader range of architectures and layers."*
> >
>
> To evaluate whether $C^{2}R$ maintains its effectiveness on larger architectures, we conducted experiments on Qwen3-8B and Llama-3-8B. The results (both at layer 20, sparsity level = 100) confirm that $C^{2}R$ successfully mitigates feature splitting and absorption at a larger scale while preserving reconstruction fidelity. In addition to the layer 12 experiments presented in the main experiments, we evaluated $C^{2}R$ on a deeper layer (layer 20) of Gemma-2-2B to verify cross-layer consistency.
>
> ***Qwen3-8B layer 20 sparsity level=100***
>
> | Method | **KL Score↑** | **Interp↑** | **Composition↓** | **Absorb↓** | **Split↓** |
> | --- | --- | --- | --- | --- | --- |
> | Batch TopK | **0.9889** | 0.922 | 0.2677 | 0.0074 | **1.1154** |
> | Matryoshka | 0.987 | *0.9277* | 0.1422 | *0.0032* | **1.1154** |
> | Ort | *0.9886* | 0.9214 | *0.0945* | 0.0034 | **1.1154** |
> | $C^{2}R$ | 0.9874 | **0.9333** | **0.0675** | **0.0018** | **1.1154** |
>
> ***Llama-3-8B layer 20 sparsity level=100***
>
> | Method | **KL Score↑** | **Interp↑** | **Composition↓** | **Absorb↓** | **Split↓** |
> | --- | --- | --- | --- | --- | --- |
> | Batch TopK | **0.986** | 0.9405 | 0.3334 | 0.147 | 1.1923 |
> | Matryoshka | 0.9853 | **0.9552** | 0.1803 | **0.0349** | *1.1154* |
> | Ort | *0.9859* | 0.9265 | *0.12* | 0.0979 | **1.0385** |
> | $C^{2}R$ | 0.9858 | *0.9443* | **0.0816** | *0.041* | **1.0385** |
>
>
>
> ***Gemma-2-2B layer 20***
>
> | Method | **KL Score↑** | **Interp↑** | **Composition↓** | **Absorb↓** | **Split↓** |
> | --- | --- | --- | --- | --- | --- |
> | TopK | 0.972 | 0.9307 | 0.327 | 0.1325 | 1.5769 |
> | Batch TopK | 0.9706 | 0.9323 | 0.3329 | 0.1087 | 1.3462 |
> | Matryoshka | **0.9727** | 0.941 | 0.1781 | **0.0205** | **1.2308** |
> | Ort | 0.9722 | *0.9543* | *0.1* | 0.0352 | *1.2692* |
> | $C^{2}R$ | *0.9727* | **0.957** | **0.0781** | *0.026* | *1.2692* |
>
> **W2: Scaling to Larger Training Corpora**
>
> > *"The SAE training setup uses only a 500M-token subset of OpenWebText... how sensitive the results are to the choice, scale, and composition of the training data."*
> >
>
> Regarding the choice and scale of the corpus, OpenWebText is a high-quality web dataset originally used for GPT-2. Contemporary works [1, 2] have utilized OpenWebText subsets ranging from 300M to 820M tokens. Our 500M-token training setup is consistent with the experimental setting of OrtSAE. We extended the training data to 1B tokens to further verify the robustness of our results. As shown in the table below, $C^{2}R$ consistently achieving optimal or near-optimal structural feature metrics without degrading reconstruction fidelity.
>
> ***1B tokens OpenWebText***
>
> | Method | **KL Score↑** | **Interp↑** | **Composition↓** | **Absorb↓** | **Split↓** |
> | --- | --- | --- | --- | --- | --- |
> | TopK | **0.9631** | 0.9196 | 0.3416 | 0.2987 | 1.2692 |
> | Batch TopK | 0.9621 | **0.9313** | 0.343 | 0.2149 | 1.1538 |
> | Matryoshka | 0.9598 | 0.887 | 0.1892 | **0.0241** | *1.0769* |
> | Ort | 0.9629 | 0.9072 | *0.1053* | 0.1058 | **1.0385** |
> | $C^{2}R$ | *0.9631* | *0.9302* | **0.087** | *0.0654* | **1.0385** |
>
> Regarding data composition, we conducted additional experiments on The Pile dataset [3] at sparsity level = 100. The Pile offers a highly diverse composition, including Academic, Internet, Prose, and Dialogue domains. As demonstrated in the table below, $C^{2}R$ maintains its strong performance on this structurally complex dataset. This confirms that our method is not sensitive to the specific composition or domain distribution of the training data.
>
> ***The Pile***
>
> | Method | KL Score↑ | Interp↑ | Composition↓ | Absorb↓ | Split↓ |
> | --- | --- | --- | --- | --- | --- |
> | Batch TopK | 0.9606 | *0.9211* | 0.3227 | 0.192 | 1.1538 |
> | Matryoshka | 0.9602 | 0.9117 | 0.1781 | **0.0158** | **1.0385** |
> | Ort | *0.9611* | 0.9202 | *0.1053* | 0.0627 | *1.0769* |
> | $C^{2}R$ | **0.9618** | **0.9424** | **0.0766** | *0.0396* | **1.0385** |
>
> *References:*
>
> [1] Mudide A, Engels J, Michaud E J, et al. Efficient Dictionary Learning with Switch Sparse Autoencoders[C]//The Thirteenth International Conference on Learning Representations.
>
> [2] Leask P, Bussmann B, Pearce M T, et al. Sparse Autoencoders Do Not Find Canonical Units of Analysis[C]//The Thirteenth International Conference on Learning Representations.
>
> [3] Gao L, Biderman S, Black S, et al. The pile: An 800gb dataset of diverse text for language modeling[J]. arXiv preprint arXiv:2101.00027, 2020.
>
> We will incorporate the above discussions and results into the final version of the manuscript.

---

> > ### Author Rebuttal · Reviewer_k9Se · 2026-04-03
> >
> > Thank you to the authors for providing additional details on the follow-up experiments. After carefully reviewing the results, I would like to keep the current score unchanged.

---

> > > ### Author Response · Authors · 2026-04-06
> > >
> > > Dear Reviewer k9Se,
> > >
> > > Thank you for your review and positive feedback. We are pleased that our rebuttal addressed your concerns. Your comments helped improve our paper, and we appreciate your time and support during this process.
> > >
> > > Sincerely,
> > >
> > > Authors of Paper 34488

---

### Official Review · Reviewer_WPYq · 2026-03-14

**Soundness:** 3
**Presentation:** 4
**Significance:** 3
**Originality:** 3
**Overall Recommendation:** 5
**Confidence:** 3

**Summary:**

The authors propose that cosine-gated Minkowski regularization of batches during SAE training yields better-grouped features at competitive reconstruction quality. They (i) theoretically characterize the behavior of feature splitting under Top K, l_1 and Minkowski regularization and demonstrate that Minkowski should reduce feature splitting under mild assumptions and (ii) empirically demonstrate competive reconstruction and interpretability and strictly superior feature orthogonality for Gemma-2-2B.

**Compliance With Llm Reviewing Policy:**

Affirmed.

**Final Justification:**

The authors present a novel, theoretically principled approach to mitigate feature splitting in SAE training and demonstrate empirically that it performs at approximate parity or beyond for Gemma-2-2B, Qwen3-8B and Llama-3-8B. Their rebuttal addressed my main concerns and I've increased my evaluation accordingly.

**Key Questions For Authors:**

* **Q1:** For a selected sparsity, can you make a table to numerically compare reconstruction vs. structural feature metrics with statistical validation? At a glance, C^2R appears to clearly outperform Matryoshka, Batch Top K and Top K but Ort appears to be a close competitor — understanding the statistical robustness of these results would improve empirical soundness.
* **Q2:** Although intuitively generalizable and valuable, can you scale this to 1-2 current-generation models (even at a single sparsity level) and/or demonstrate effect on downstream tasks (circuit discovery, model steering, etc.) to concretely demonstrate significance?

**Limitations:**

Yes

**Strengths And Weaknesses:**

* **S1:** Unified geometric framework elegantly reduces splitting and absorption to a single phenomenon, with a clean Minkowski-based proof that per-sample TopK incentivize the pathology (Lemma 4.1).
* **S2:** Empirical results deliver on the theory: near-zero absorption, minimal splitting, and lowest composition — all without degrading reconstruction, interpretability, or disentanglement, unlike Matryoshka SAEs which trade fidelity for hierarchy.
* **S3:** Manuscript was well-written and easy to understand on a 1st pass — motivation, pedagogical setup to build intuition, broader theoretical intervention, empirical results.
* **W1 (statistical validity):** All comparisons are single-run point estimates with no error bars, confidence intervals, or significance tests — given apparent competitiveness of Ort vs. C^2R and apparent sensitivity of SAEs to random seeds (Paulo & Belrose, 2025), this undermines confidence in reported margins.
* **W2 (empirical relevance):** All experiments use Gemma-2-2B layer 12 on OpenWebText — the identical setup as OrtSAE and Matryoshka — so generalization to other models (e.g., Llama-3), layers, or architectures (MoE) is unproven.
* **W3 (experimental rigor):** No ablations or perturbation analyses isolate load-bearing design decisions or assumptions: ReLU cosine gate, the nearest-neighbor restriction, or $\lambda_{C^2R}$ (the hyperparameter sweep is mentioned but not shown). Eq. 13's sufficient condition is never empirically verified.
* **W4 (significance):** The entire motivation is improving causal tracing, circuit discovery, and model steering, yet no downstream task is evaluated — the paper stops at structural metrics, leaving the practical impact of reduced splitting/absorption as pure conjecture.

---

> ### Author Rebuttal · Authors · 2026-03-31
>
> Thank you for your thorough review and constructive feedback. In the tables below, the best and second-best values for each metric are highlighted in **bold** and *italics*, respectively. Here are our responses to your specific concerns:
>
> **W1 & Q1: Statistical Validity**
>
> To address the concern regarding single-run point estimates, we conducted 5 independent repeated experiments for the main results in Figures 2 and 3. We report the 95% confidence intervals (CI) below. The results validate our initial findings: C$^2$R consistently minimizes feature composition and absorption without degrading reconstruction fidelity, and the improvements are statistically robust compared to the baselines.
>
> | Method | **KL Score↑** | **Interp↑** | **Composition↓** | **Absorb↓** | **Split↓** |
> | --- | --- | --- | --- | --- | --- |
> | Batch TopK | *0.9621 ± 0.0006* | **0.9311 ± 0.0098** | 0.3325 ± 0.0003 | 0.1908 ± 0.024 | 1.0962 ± 0.0722 |
> | Matryoshka | 0.9597 ± 0.0008 | *0.9311 ± 0.012* | 0.187 ± 0.0003 | **0.036 ± 0.0066** | 1.0673 ± 0.0361 |
> | Ort | 0.9615 ± 0.0006 | 0.9297 ± 0.0082 | *0.1038 ± 0.0004* | 0.0819 ± 0.0203 | *1.0577 ± 0.0218* |
> | C2R | **0.9628 ± 0.0004** | 0.9309 ± 0.0052 | **0.0991 ± 0.0003** | *0.0656 ± 0.0185* | **1.0385 ± 0.0238** |
>
> **W2 & Q2: Scale Experiments to Larger Models**
>
> To verify generalization, we scaled our experiments to Qwen3-8B and Llama-3-8B (both at layer 20, sparsity level = 100). The results confirm that C$^2$R maintains its effectiveness on larger, current-generation models, achieving optimal or near-optimal performance on structural feature metrics while preserving reconstruction fidelity.
>
> ***Qwen3-8B layer 20 sparsity level=100***
>
> | Method | **KL Score↑** | **Interp↑** | **Composition↓** | **Absorb↓** | **Split↓** |
> | --- | --- | --- | --- | --- | --- |
> | Batch TopK | **0.9889** | 0.922 | 0.2677 | 0.0074 | **1.1154** |
> | Matryoshka | 0.987 | *0.9277* | 0.1422 | *0.0032* | **1.1154** |
> | Ort | *0.9886* | 0.9214 | *0.0945* | 0.0034 | **1.1154** |
> | $C^{2}R$ | 0.9874 | **0.9333** | **0.0675** | **0.0018** | **1.1154** |
>
> ***Llama-3-8B layer 20 sparsity level=100***
>
> | Method | **KL Score↑** | **Interp↑** | **Composition↓** | **Absorb↓** | **Split↓** |
> | --- | --- | --- | --- | --- | --- |
> | Batch TopK | **0.986** | 0.9405 | 0.3334 | 0.147 | 1.1923 |
> | Matryoshka | 0.9853 | **0.9552** | 0.1803 | **0.0349** | *1.1154* |
> | Ort | *0.9859* | 0.9265 | *0.12* | 0.0979 | **1.0385** |
> | $C^{2}R$ | 0.9858 | *0.9443* | **0.0816** | *0.041* | **1.0385** |
>
> **W3: Ablations & Hyperparameters**
>
> **Ablation Study:** We evaluated the removal of the nearest-neighbor restriction (NoNNR) and the ReLU cosine gate (NoRCG). Removing either component imposes an overly aggressive merging constraint. While this severely penalizes feature absorption, it does so at the unacceptable cost of significantly degrading reconstruction ability (e.g., KL Score drops to 0.8439 without RCG). This demonstrates that the cautious, gated constraints of C$^2$R are necessary to mitigate feature pathologies without destroying dictionary utility.
>
> | Method | **KL Score↑** | **Interp↑** | **Composition↓** | **Absorb↓** | **Split↓** |
> | --- | --- | --- | --- | --- | --- |
> | Batch TopK | 0.9598 | 0.9208 | 0.3321 | 0.1985 | *1.1154* |
> | ort | *0.9617* | 0.9208 | *0.1046* | 0.0606 | *1.1154* |
> | C2R NoNNR | 0.9439 | *0.9488* | 0.5655 | **0.0255** | 1.1538 |
> | C2R NoRCG | 0.8439 | **0.9873** | 0.4326 | *0.052* | 1.1923 |
> | C2R | **0.9629** | 0.9239 | **0.099** | 0.059 | **1.0769** |
>
> **Hyperparameter Sensitivity ($\lambda_{C^2R}$):** We selected $\lambda_{C^2R} = 5$ as it maximizes the reduction in feature composition and absorption without sacrificing reconstruction fidelity.
>
> | Method | **KL Score↑** | **Interp↑** | **Composition↓** | **Absorb↓** | **Split↓** |
> | --- | --- | --- | --- | --- | --- |
> | Batch TopK | 0.9598 | 0.9196 | 0.3321 | 0.1985 | 1.1154 |
> | ort | 0.9617 | 0.9196 | 0.1046 | 0.0606 | 1.1154 |
> | lambda_C2R=0.1 | 0.9629 | 0.9291 | 0.2944 | 0.1926 | **1.0385** |
> | lambda_C2R=0.5 | **0.9638** | **0.9496** | 0.2138 | 0.1726 | **1.0385** |
> | lambda_C2R=1 | *0.9633* | *0.9407* | 0.1676 | 0.1349 | 1.1154 |
> | lambda_C2R=5 | 0.9629 | 0.9239 | *0.099* | *0.059* | *1.0769* |
> | lambda_C2R=10 | 0.9521 | 0.9168 | **0.0777** | **0.0425** | **1.0385** |
>
> **W4 & Q2: Downstream Tasks**
>
> We evaluate the practical utility of SAEs on two downstream causal intervention tasks: Spurious Correlation Removal (SCR) and Targeted Probe Perturbation (TPP). As shown below, C$^2$R substantially outperforms TopK, Batch TopK, and Ort, while remaining highly competitive with Matryoshka.
>
> | Method | **TopK** | **Batch TopK** | **Matryoshka** | **ort** | **C2R** |
> | --- | --- | --- | --- | --- | --- |
> | SCR↑ | 0.0481 | 0.0353 | **0.1172** | 0.0859 | *0.1047* |
> | TPP↑ | 0.0038 | 0.0045 | **0.037** | 0.0061 | *0.0226* |
>
> We will incorporate the above discussions and results into the final version of the manuscript.

---

> > ### Author Rebuttal · Reviewer_WPYq · 2026-04-03
> >
> > I appreciate the authors constructive engagement and thorough response. I've updated my score accordingly. Great work!

---

> > > ### Author Response · Authors · 2026-04-06
> > >
> > > Dear Reviewer WPYq,
> > >
> > > Thank you for your review and positive feedback. We are pleased that our rebuttal addressed your concerns. Your comments helped improve our paper, and we appreciate your time and support during this process.
> > >
> > > Sincerely,
> > >
> > > Authors of Paper 34488

---

### Decision · Program_Chairs · 2026-04-30

**Decision:**

Accept (regular)

**Comment:**

- The reviews broadly agree that the paper makes a novel and well-motivated contribution by introducing a principled method, supported by solid theory and strong empirical results on the paper’s main metrics.
- The main initial concerns were about limited scope, missing statistical validation, lack of ablations, and the absence of downstream evidence, but these were largely addressed in the rebuttal through additional experiments.
- As a result, the discussion shifted clearly in favor of the paper, with reviewers stating that their major concerns had been resolved and one reviewer explicitly raising their score to accept.
- Open points are relatively minor and primarily concern clarifications and additions for the final version rather than fundamental flaws in the work.